# PRINCIPLED LATENT DIFFUSION FOR GRAPHS VIA LAPLACIAN AUTOENCODERS

## ABSTRACT

Graph diffusion models achieve state-of-the-art performance in graph generation but suffer from quadratic complexity in the number of nodes—and much of their capacity is wasted modeling the absence of edges in sparse graphs. Inspired by latent diffusion in other modalities, a natural idea is to compress graphs into a low-dimensional latent space and perform diffusion there. However, unlike images or text, graph generation requires nearly lossless reconstruction, as even a single error in decoding an adjacency matrix can render the entire sample invalid. This challenge has remained largely unaddressed. We propose LG-Flow, a latent graph diffusion framework that directly overcomes these obstacles. A permutation-equivariant autoencoder maps each node into a fixed-dimensional embedding from which the full adjacency is provably recoverable, enabling near-lossless reconstruction for both undirected graphs and DAGs. The size of this latent representation scales linearly with the number of nodes, eliminating the quadratic bottleneck and making it feasible to train larger and more expressive models. In this latent space, we train a Diffusion Transformer with flow matching, enabling efficient and expressive graph generation. Our approach achieves competitive results against state-of-the-art graph diffusion models, while achieving up to $1000\times$ speed-up.

## 1 INTRODUCTION

Generative modeling of graphs has advanced rapidly in recent years, with diffusion models achieving state-of-the-art performance across various domains, including molecules (Igashov et al., 2024), combinatorial optimization (Sun and Yang, 2023), and neural architecture search. However, most existing graph diffusion models operate directly in graph space, which entails quadratic computational complexity in the number of nodes (Vignac et al., 2023). These models typically learn both node and edge representations, since predicting edges solely from node features has proven insufficient across benchmarks (Qin et al., 2023). As a result, scalability remains limited. Quadratic complexity also leads to disproportionate memory consumption relative to the number of parameters, which restricts model capacity compared to image-based diffusion. In addition, attempts to adapt scalable auto-regressive sequence models to graphs break permutation equivariance when graphs are linearized, e.g., Chen et al. (2025), which limits their applicability.

In contrast, latent diffusion has advanced image generation by shifting the generative process into a compressed latent space (Esser et al., 2024; Rombach et al., 2022). A VAE maps data into low-dimensional representations, where Gaussian diffusion or flow models operate more efficiently, avoiding wasted capacity on imperceptible details. This yields efficient training, faster inference, and supports high-capacity architectures.

Adapting this principle to graphs is highly appealing but presents unique challenges. Unlike in images, where minor decoding errors are visually negligible, a single mistake in reconstructing an adjacency matrix can destroy structural validity—for instance, by deleting a chemical bond or misclassifying its type. Compression must therefore not come at the expense of reconstruction fidelity. The decoder must achieve near-lossless reconstruction to guarantee valid samples. Unfortunately, graph autoencoders are rarely evaluated for reconstruction quality (Kipf and Welling, 2016; Lee and Min, 2022; Li et al., 2020), with the notable exception of Boget et al. (2024).

Despite these obstacles, the latent diffusion paradigm is particularly promising for graphs. Most real-world graphs are sparse (see Table 1), which means diffusion models trained in graph space devote most of their capacity to modeling the absence of edges. A latent approach can alleviate this inefficiency,

reduce quadratic complexity, and enable the training of larger and more expressive generative models. This motivates designing latent graph representations whose size—understood as the total number of dimensions—scales linearly with the number of nodes, for example, by assigning each node a fixed-dimensional latent vector.

Latent diffusion offers modularity and reusability: instead of designing graph-specific diffusion models, one can employ generic denoising architectures, reusing methods and optimization techniques from the thriving image generation community such as the *Diffusion Transformer* (DiT) (Peebles and Xie, 2023). Once the encoder is trained, the diffusion stage is uniform across modalities (see, e.g., Joshi et al. (2025)). In the graph domain, this approach unifies undirected graph (Qin et al., 2025; Vignac et al., 2023) and *directed acyclic graph* (DAG) generation (Carballo-Castro et al., 2025; Li et al., 2025a), which are typically treated separately. The latter is especially relevant to highly impactful applications like chip design and circuit synthesis (Mishchenko and Miyasaka, 2023; Li et al., 2025b; Phothilimthana et al., 2023b). This modular view shifts the focus to designing strong, cheaper-to-train graph autoencoders, which rely on linear-complexity *(Message-Passing) Graph Neural Networks* (MPNNs) (Gilmer et al., 2017; Scarselli et al., 2009).

Table 1: Average density of training graphs across common benchmarks.

| Dataset | Density |
|---------|---------|
| Planar | 0.09 |
| Tree | 0.03 |
| Ego | 0.04 |
| Moses | 0.11 |
| Guacamol | 0.09 |
| TPU Tile | 0.06 |

From these considerations, we derive four requirements for a principled latent graph diffusion model. **(R1)** The autoencoder must provide provable reconstruction guarantees that ensure the structural validity of decoded graphs. **(R2)** The size of a graphs' latent representations should scale linearly with the number of nodes, enabling scalability to large graphs. **(R3)** Graph-specific inductive biases should be encapsulated in the autoencoder, allowing the diffusion model itself to remain generic. **(R4)** The overall framework should maintain competitive generative performance while avoiding the quadratic complexity of graph-space diffusion.

However, as argued above, existing latent graph diffusion models fail to satisfy all four requirements. Most lack reconstruction guarantees (Yang et al., 2024b; Fu et al., 2024), while others forfeit efficiency by relying on ad-hoc transformer designs (Nguyen et al., 2024) or a quadratic number of edge tokens (Zhou et al., 2024).

**Present work** Hence, to address the above requirements **R1**-**R4**, we propose a latent graph diffusion framework. Concretely, we

1. derive the *Laplacian Graph Variational Autoencoder* (LG-VAE), a principled permutation-equivariant graph autoencoder with provable reconstruction guarantees, extending theory from positional encodings in *Graph Transformers* (GTs) to both undirected graphs and DAGs.

2. Our autoencoder maps each node to a fixed-dimensional embedding from which the adjacency matrix is recoverable, yielding compact latent representations that scale linearly with graph size.

3. In this latent space, we apply flow matching with a DiT architecture, achieving competitive performance across benchmarks while maintaining scalability and architectural simplicity. Our approach (LG-Flow) delivers competitive results on both synthetic and real-world benchmarks, while achieving speed-ups ranging from $10\times$ to $1000\times$.

*Overall, our LG-Flow enables diffusion-based graph generative models to scale to larger graph sizes with significantly improved inference efficiency, paving the way for their application to domains previously out of reach.*

## 2 RELATED WORK

Here, we discuss related work. A more detailed discussion is provided in Appendix A, including an overview of prior graph diffusion models.

**Graph generation** Graph generation methods are typically categorized into two main groups. *Autoregressive models* build graphs by progressively adding nodes and edges (You et al., 2018; Jang et al., 2024). Their main advantage is computational efficiency, since they do not need to materialize the full adjacency matrix. However, they depend on an arbitrary node ordering to transform the graph into a

sequence, either learned or defined through complex algorithms, which breaks permutation equivariance. In contrast, *one-shot models* generate the entire graph at once, thereby preserving permutation equivariance. Many approaches have been explored in this class, starting with GVAEs (Kipf and Welling, 2016) and GANs (Martinkus et al., 2022). Following their success in other modalities such as images, diffusion models (Ho et al., 2020; Song et al., 2021; Lipman et al., 2023) quickly emerged as the dominant approach for graph generation.

**Latent diffusion models** Latent diffusion models (LDMs) (Rombach et al., 2022) generate data by applying the diffusion process in a compressed latent space rather than pixel space, typically using a VAE for encoding and decoding. This reduces computation while preserving detail, and underlies models such as Stable Diffusion. More recently, DiTs (Peebles and Xie, 2023) emerged as effective denoisers across modalities (Esser et al., 2024; Joshi et al., 2025). For graphs, however, latent diffusion has not matched the performance of discrete diffusion models (Fu et al., 2024; Nguyen et al., 2024; Yang et al., 2024a). Only Nguyen et al. (2024) evaluate autoencoder reconstruction, and only on small QM9 graphs, with an ad-hoc transformer that materializes the full attention matrix, reducing efficiency. Similarly, Zhou et al. (2024) achieve competitive results on MOSES but rely on $n^2$ edge tokens from the full adjacency matrix, also limiting efficiency. Other works target only molecule generation (Ketata et al., 2025; Wang et al., 2024; Bian et al., 2024) or tasks like link prediction (Fu et al., 2024) and regression (Zhou et al., 2024). In contrast, our approach ensures the adjacency matrix is provably recoverable from the latent space.

## 3 BACKGROUND

In this section, we introduce theoretical tools used throughout the paper, including graphs, adjacency-identifying positional encodings, and Laplacian positional encodings. For completeness, the formal introduction of flow matching is deferred to Appendix B. Note that since flow matching and Gaussian diffusion are essentially equivalent, we use the two terms interchangeably (Gao et al., 2024).

**Notations** Let $\mathbb{N} := \{1, 2, \dots\}$ and for $n \in \mathbb{N}$ define $[n] := \{1, \dots, n\} \subset \mathbb{N}$. We denote by $\mathbb{R}^{+*}$ the set of strictly positive real numbers. We consider *node- and edge-labeled undirected graphs* $G$, i.e., tuples $(V(G), E(G), \ell_x, \ell_e)$ with up to $n$ nodes and $m$ edges, without self-loops or isolated nodes. Here, $V(G)$ is the set of nodes, $E(G)$ the set of edges, $\ell_x \colon V(G) \to [a]$, for $a \in \mathbb{N}$, assigns each node one of $a$ discrete labels, and $\ell_e \colon E(G) \to [b]$, for $b \in \mathbb{N}$, assigns each edge one of $b$ discrete labels. An undirected edge $\{u, v\} \in E(G)$ is written either $(u, v)$ or $(v, u)$. We also consider *node- and edge-labeled directed acyclic graphs* (DAGs), defined analogously except that $(u, v)$ denotes a directed edge from $u$ to $v$ and the graph contains no directed cycles. Throughout, we assume an arbitrary but fixed ordering of nodes so that each node corresponds to a number in $[n]$. The adjacency matrix of $G$ is denoted $\boldsymbol{A}(G) \in \{0, 1\}^{n \times n}$, where $\boldsymbol{A}(G)_{ij} = 1$ if and only if nodes $i$ and $j$ are connected. We construct a node feature vector $\boldsymbol{X} \in [a]^n$ consistent with $\ell_x$, so that $X_i = X_j$ if and only if $\ell_x(i) = \ell_x(j)$. Similarly, we define an edge feature matrix $\boldsymbol{E} \in [b]^m$ consistent with $\boldsymbol{A}$ and $\ell_e$, so that $E_{ij} = E_{kl}$ if and only if $\ell_e((i, j)) = \ell_e((k, l))$. For any matrix $\boldsymbol{M}$, we denote by $\boldsymbol{M}_i$ its $i$-th row. For a real-valued matrix $\boldsymbol{M} \in \mathbb{R}^{n \times n}$, $\boldsymbol{M}^\top$ denotes its transpose, and for a complex matrix $\boldsymbol{N}$, $\boldsymbol{N}^*$ denotes its Hermitian transpose. For two binary matrices $\boldsymbol{P}$ and $\boldsymbol{Q}$, we denote their logical disjunction $\boldsymbol{P} \vee \boldsymbol{Q}$ where $(\boldsymbol{P} \vee \boldsymbol{Q})_{ij} = 1$ if $\boldsymbol{P}_{ij} = 1$ or $\boldsymbol{Q}_{ij} = 1$, and 0 otherwise. Let $\boldsymbol{u}$ and $\boldsymbol{v}$ be two vectors; $\boldsymbol{u} \odot \boldsymbol{v}$ denotes the element-wise product between them. Finally, the *graph Laplacian* is $\boldsymbol{L} := \boldsymbol{D} - \boldsymbol{A}$, where $\boldsymbol{D} \in \mathbb{N}^{n \times n}$ is the degree matrix. Its eigenvalue decomposition is $\boldsymbol{L} := \boldsymbol{U} \boldsymbol{\Lambda} \boldsymbol{U}^\top$, where $\boldsymbol{\Lambda} := (\lambda_1, \dots, \lambda_n)$ is the vector of eigenvalues (possibly repeated) and $\boldsymbol{U} \in \mathbb{R}^{n \times n}$ the matrix of eigenvectors, with the $i$-th column $\boldsymbol{U}_i^\top$ corresponding to eigenvalue $\lambda_i$. We denote by $\boldsymbol{I}_d$ the $d$-dimensional identity matrix.

### 3.1 ADJACENCY-IDENTIFYING POSITIONAL ENCODINGS

The design of our autoencoder extends *adjacency-identifying positional encodings*, introduced in the literature on positional encodings for GTs (Müller and Morris, 2024). Intuitively, such encodings guarantee that the adjacency matrix of an *undirected* graph can be reconstructed by the attention mechanism from node embeddings, provided the positional encoding is chosen appropriately.

Formally, let $G$ be a directed graph $(V(G), E(G))$ with $n$ nodes, and let $\boldsymbol{P} \in \mathbb{R}^{n \times d}$ denote a matrix of $d$-dimensional node embeddings. Define

$$\tilde{\boldsymbol{P}} \coloneqq \frac{1}{\sqrt{d}} \boldsymbol{P} \boldsymbol{W}^Q \left( \boldsymbol{P} \boldsymbol{W}^K \right)^\top, \tag{1}$$

where $\boldsymbol{W}^Q, \boldsymbol{W}^K \in \mathbb{R}^{d \times d}$. We say that $\boldsymbol{P}$ is *adjacency-identifying* if there exist $\boldsymbol{W}^Q, \boldsymbol{W}^K$ such that for every row $i$ and column $j$, $\tilde{\boldsymbol{P}}_{ij} = \max_k \tilde{\boldsymbol{P}}_{ik} \iff \boldsymbol{A}(G)_{ij} = 1$, where $\boldsymbol{A}(G)$ denotes the adjacency matrix of $G$. Note that in connected graphs, each row has between one and $(n-1)$ maxima. We further say that a matrix $\boldsymbol{Q} \in \mathbb{R}^{n \times d}$ is *sufficiently adjacency-identifying* if it approximates an adjacency-identifying matrix $\boldsymbol{P}$ to arbitrary precision, i.e., $\|\boldsymbol{P} - \boldsymbol{Q}\|_{\mathrm{F}} < \epsilon$, for all $\epsilon > 0$.

**LPE** The *Laplacian Positional Encoding* (LPE), introduced in Dwivedi et al. (2022); Müller and Morris (2024), is an example of a sufficiently adjacency-identifying positional encoding, constructed as follows. Let $\epsilon \in \mathbb{R}^l$ be a learnable vector initialized to zero. Let $\phi \colon \mathbb{R}^2 \to \mathbb{R}^d$ denote a feedforward network, applied row-wise, and let $\rho \colon \mathbb{R}^{n \times d} \to \mathbb{R}^d$ denote a permutation-equivariant neural network applied row-wise. For an eigenvector matrix $\boldsymbol{U}$ of the graph Laplacian and its corresponding eigenvalues $\boldsymbol{\Lambda}$, we define

$$\phi(\boldsymbol{U}, \boldsymbol{\Lambda}) \coloneqq \left[ \phi\left(\boldsymbol{U}_1 \,\|\, \boldsymbol{\Lambda} + \epsilon\right), \dots, \phi\left(\boldsymbol{U}_n \,\|\, \boldsymbol{\Lambda} + \epsilon\right) \right] \in \mathbb{R}^{n \times n \times d}, \tag{2}$$

where $\boldsymbol{U}_i \,\|\, \boldsymbol{\Lambda} + \epsilon \in \mathbb{R}^{n \times 2}$ denotes row-wise concatenation. The network $\phi$ is applied on the last dimension, which yields $\phi(\boldsymbol{U}_i \,\|\, \boldsymbol{\Lambda} + \epsilon) \in \mathbb{R}^{n \times d}$, for each $i \in [n]$. Stacking these outputs over all nodes produces $\phi(\boldsymbol{U}, \boldsymbol{\Lambda})$.

The LPE is then obtained by aggregating contributions across all eigenvectors and eigenvalues. Specifically,

$$\mathrm{LPE}(\boldsymbol{U}, \boldsymbol{\Lambda}) \coloneqq \rho\left( \sum_{i=1}^{n} \phi(\boldsymbol{U}, \boldsymbol{\Lambda})_i^\top \right), \tag{3}$$

where $\phi(\boldsymbol{U}, \boldsymbol{\Lambda})_i^\top \in \mathbb{R}^{n \times d}$ denotes the $i$-th column of $\phi(\boldsymbol{U}, \boldsymbol{\Lambda}) \in \mathbb{R}^{n \times n \times d}$. Intuitively, this means that for each node we combine the information from all Laplacian (linearly independent) eigenvectors and eigenvalues, and then apply a DeepSet (Zaheer et al., 2017) over the (multi)set $\left\{\!\!\left\{ \phi(\boldsymbol{U}_i \,\|\, \boldsymbol{\Lambda} + \epsilon) \right\}\!\!\right\}_{i=1}^n$.

## 4 PRINCIPLED LATENT DIFFUSION FOR GRAPHS

We now present our latent diffusion framework for graphs. The core idea is to design an autoencoder with provable reconstruction guarantees (see Figure 1 for an overview), covering both undirected graphs and DAGs, and train a diffusion model in the resulting latent space. An overview of the whole framework is provided in Figure 2.

### 4.1 LAPLACIAN GRAPH VARIATIONAL AUTOENCODER

In latent diffusion models, the expressive power of the decoder sets an upper bound on the quality of generated samples. For graphs, even a single incorrect edge can invalidate the entire structure, e.g., by breaking chemical validity in molecules, so strong reconstruction guarantees are essential. However, most existing graph autoencoders are not benchmarked for reconstruction accuracy, raising concerns about their reliability (Fu et al., 2024; Nguyen et al., 2024).

We address this by building on LPEs; see Section 3.1. In fact, we require node embeddings that are expressive enough to preserve all structural information, so that the adjacency matrix can be provably recovered from them. The LPE is a natural choice: by construction, it yields embeddings that are *sufficiently adjacency-identifying*, i.e., they preserve all information necessary for reconstructing the adjacency matrix. This property also provides a natural recipe for the decoder: adjacency can be recovered through a bilinear layer (as in Equation (1)), followed by a row-wise argmax. This design satisfies two of our requirements: (**R1**) provable reconstruction guarantees and (**R2**) graph latent representations size that scales linearly with the number of nodes.

Building on this foundation, we introduce the *Laplacian Graph Variational Autoencoder* (LG-VAE). Following the VAE framework, the encoder $\mathcal{E}$ maps a graph $G$ with node features $\boldsymbol{X} \in \mathbb{R}^n$, edge features $\boldsymbol{E} \in \mathbb{R}^m$, and Laplacian $\boldsymbol{L} = \boldsymbol{U}\boldsymbol{\Lambda}\boldsymbol{U}^\top$ to the parameters of a Gaussian posterior with mean $\boldsymbol{\mu_Z}$ and log-variance $\log \boldsymbol{\sigma_Z}^2$. Latents $\boldsymbol{Z} \in \mathbb{R}^{n \times d}$ are then sampled using the reparameterization trick and passed to the decoder $\mathcal{D}$, which reconstructs $\hat{G} = \mathcal{D}(\boldsymbol{Z})$; see Figure 1 for an overview.

Figure 1: Overview of the LG-VAE. The encoder $\mathcal{E}$ (left) encodes structure via $\phi$ and node/edge labels, then aggregates them with $\rho$. Latents $\boldsymbol{Z}$ are sampled using the reparameterization trick and passed to the decoder $\mathcal{D}$ (right). Node labels are decoded directly, while adjacency is decoded by (a) computing scores via bilinear dot products for $b + 1$ heads, (b) processing scores with a row-wise DeepSet, and (c) concatenating outputs across heads. The final adjacency $\hat{\boldsymbol{A}}$ is obtained with an argmax.

**Encoder**   The encoder extends the LPE to incorporate node and edge labels. We compute

$$\boldsymbol{H}_\phi := \sum_{i=1}^n \phi(\boldsymbol{U}, \boldsymbol{\Lambda})_i^\top \in \mathbb{R}^{n \times d}, \quad \boldsymbol{H}_X := \boldsymbol{X}\boldsymbol{W}^X + \boldsymbol{b}^X \in \mathbb{R}^{n \times d}, \quad \boldsymbol{H}_E := \boldsymbol{E}\boldsymbol{W}^E + \boldsymbol{b}^E \in \mathbb{R}^{m \times d},$$

where $\boldsymbol{W}^X, \boldsymbol{W}^E$ are projection matrices, and $\boldsymbol{b}^X, \boldsymbol{b}^E$ the associated bias, and feed them into $\rho$, yielding

$$\boldsymbol{\mu_Z}, \log \boldsymbol{\sigma_Z}^2 := \mathcal{E}(G, \boldsymbol{X}, \boldsymbol{E}, \boldsymbol{U}, \boldsymbol{\Lambda}) = \rho(\boldsymbol{H}_X + \boldsymbol{H}_\phi, \boldsymbol{H}_E). \tag{4}$$

Here $\boldsymbol{H}_\phi$ is treated as an additional node feature, and $\rho$ can be any permutation-equivariant network. In our experiments, we use GIN (Xu et al., 2019) and GINE (Hu et al., 2020). A latent sample is then obtained as

$$\boldsymbol{Z} = \boldsymbol{\mu_Z} + \boldsymbol{\sigma_Z} \odot \epsilon, \qquad \epsilon \sim \mathcal{N}(0, \boldsymbol{I}_d).$$

Note that, as detailed in Appendix D.1, and depicted in Figure 1, we only use the eigenvectors associated to the $k$ lowest eigenvalues, as using the full matrix eigenvectors might be impractical for large graphs.

**Decoder**   The decoder reconstructs both node and edge labels as well as the adjacency matrix. Node labels are predicted as

$$\hat{\boldsymbol{X}} := \mathcal{D}_x(\boldsymbol{Z}) = \mathrm{softmax}(\boldsymbol{Z}\boldsymbol{W}^{D_x} + \boldsymbol{b}^{D_x}), \quad \hat{\boldsymbol{X}} \in \mathbb{R}^{n \times a},$$

where $\boldsymbol{W}^{D_x} \in \mathbb{R}^{d \times a}$ is a projection matrix and $\boldsymbol{b}^{D_x}$ is a bias. To reconstruct the adjacency matrix, we first process $\boldsymbol{Z}$ to produce bilinear scores,

$$\tilde{\boldsymbol{Z}} := \tfrac{1}{\sqrt{d}} \boldsymbol{Z}\boldsymbol{W}^Q (\boldsymbol{Z}\boldsymbol{W}^K)^\top \in \mathbb{R}^{n \times n}.$$

In principle, because the LPE is sufficiently adjacency-identifying, we should be able to detect edges by looking at multiple maxima over the rows of $\tilde{\boldsymbol{Z}}$, which is impractical. Instead, we treat the detection of maxima—and thus of edges—as a binary classification problem. Since rows may have variable lengths, as graphs have varying sizes, and since we need to preserve permutation equivariance, we instantiate this classifier as a DeepSet. It takes the rows of $\tilde{\boldsymbol{Z}}$ as inputs and outputs individual logits for each $\hat{\boldsymbol{A}}_{ij}$. Formally, our decoder can be written as

$$\hat{\boldsymbol{A}} = \mathcal{D}_e(\boldsymbol{Z}) = \sigma\left(\mathrm{DeepSet}\left(\tilde{\boldsymbol{Z}}\right)\right), \quad \hat{\boldsymbol{A}} \in \mathbb{R}^{n \times n},$$

where $\mathrm{DeepSet} \colon \mathbb{R}^n \to \mathbb{R}^n$ is applied row-wise and $\sigma(\cdot)$ is a point-wise applied sigmoidal function. To decode edge labels along with the graph structure, we extend the bilinear layer to a multi-headed version; see Appendix B.1 for details.

**Loss function**   Following standard practice in VAEs, the training objective combines node and edge reconstruction losses, with KL regularization:

$$\mathcal{L}(\hat{G}) = \mathcal{L}_{\mathrm{node}}(\boldsymbol{X}, \hat{\boldsymbol{X}}) + \mathcal{L}_{\mathrm{edge}}(\boldsymbol{A}, \hat{\boldsymbol{A}}) + \beta\, \mathcal{L}_{\mathrm{KL}}(\mu_{\boldsymbol{Z}}, \sigma_{\boldsymbol{Z}}), \quad \beta > 0,$$

where $\mathcal{L}_{\mathrm{node}}(\boldsymbol{X}, \hat{\boldsymbol{X}}) = \mathrm{CrossEntropy}(\boldsymbol{X}, \hat{\boldsymbol{X}}), \mathcal{L}_{\mathrm{edge}}(\boldsymbol{A}, \hat{\boldsymbol{A}}) = \mathrm{CrossEntropy}(\boldsymbol{A}, \hat{\boldsymbol{A}})$, and $\mathcal{L}_{\mathrm{KL}}(\mu_{\boldsymbol{Z}}, \sigma_{\boldsymbol{Z}}) = D_{\mathrm{KL}}\big(\mathcal{N}(\boldsymbol{Z}; \mu_{\boldsymbol{Z}}, \sigma_{\boldsymbol{Z}}) \,\|\, \mathcal{N}(0, \boldsymbol{I}_d)\big)$. Note that the decoder can be readily adapted to continuous features by outputting a scalar instead of a categorical vector and replacing the cross-entropy loss with a standard mean-squared error.

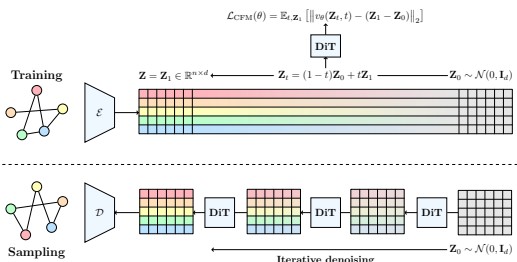

Figure 2: Overview of LG-Flow. During training (top), the frozen encoder maps graphs into latent representations $\boldsymbol{Z}$. Noisy latents $\boldsymbol{Z}_t$ are sampled along a linear interpolation path and passed through the DiT, which predicts $v_{\boldsymbol{\theta}}$ and is optimized with the conditional flow matching loss. During sampling (bottom), noise is generated and iteratively denoised using the trained DiT. The final latents are then decoded into synthetic graphs using the frozen decoder.

## 4.2 LATENT FLOW MATCHING

Once the LG-VAE is trained, we freeze its parameters and train a generative model in the latent space of the LG-VAE. We adopt FM (Lipman et al., 2023), implemented with a DiT (Peebles and Xie, 2023) and name our latent flow matching model LG-Flow. Given latents $\boldsymbol{Z} \in \mathbb{R}^{n \times d}$, we sample $t \in [0, 1]$ and from $\boldsymbol{Z}_t = (1-t)\boldsymbol{Z} + t\boldsymbol{Z}_0$ with $\boldsymbol{Z}_0 \sim \mathcal{N}(0, \boldsymbol{I}_d)$. We follow the logit-normal time distribution (Esser et al., 2024), which emphasizes intermediate timesteps, and optimize the FM objective

$$\mathcal{L}_{\text{CFM}}(\theta) = \mathbb{E}_{t, \boldsymbol{Z}} \left[ \left\| v_{\boldsymbol{\theta}}(\boldsymbol{Z}_t, t) - (\boldsymbol{Z} - \boldsymbol{Z}_0) \right\|_2 \right].$$

This latent formulation yields several benefits. First, training is efficient since DiT operates on low-dimensional node embeddings rather than directly predicting the full adjacency matrix. While the Transformer architecture has quadratic complexity in theory, efficient implementations yield near-linear scaling. Consequently, the only step with quadratic complexity is decoding, handled by our lightweight decoder. Secondly, using a modality-agnostic FM backbone highlights modularity, i.e., the encoder handles all graph-specific complexity, while the diffusion stage remains generic.

## 4.3 EXTENDING TO DAGs

The LG-VAE described so far applies to undirected graphs. The central role of DAGs in applications such as chip design and logical circuit synthesis motivates extending the LG-VAE and LG-Flow framework beyond undirected graphs to the directed setting. Extending it to DAGs is non-trivial, since the LPE relies on the eigenvalue decomposition of the symmetric Laplacian, which discards edge directionality when applied to directed graphs. To preserve this information, we adopt the *magnetic Laplacian* (Forman, 1993; Shubin, 1994; Colin de Verdière, 2013; Furutani et al., 2019; Geisler et al., 2023), a Hermitian matrix that generalizes the Laplacian to directed graphs. It is defined as $\boldsymbol{L}_q := \boldsymbol{D}_s - \boldsymbol{A}_s \odot e^{i\Theta^{(q)}}$, where $\boldsymbol{A}_s := \boldsymbol{A} \vee \boldsymbol{A}^\top$ is the symmetrized adjacency, $\boldsymbol{D}_s$ its degree matrix, $\Theta^{(q)} = 2\pi q(A_{ij} - A_{ji})$, and $q \in \mathbb{R}^{+*}$ is a fixed real parameter. This matrix admits an eigendecomposition $\boldsymbol{L}_q = \boldsymbol{U}\boldsymbol{\Lambda}\boldsymbol{U}^*$ with eigenvectors in $\mathbb{C}^n$, which we use to introduce the *magnetic LPE* (mLPE).

The mLPE mirrors the construction of the LPE but separates the real and imaginary parts of the eigenvectors, allowing the use of standard real-valued neural networks. Concretely, let $\boldsymbol{U}^R = \Re(\boldsymbol{U})$ and $\boldsymbol{U}^I = \Im(\boldsymbol{U})$ be the real and imaginary part of $\boldsymbol{U}$, respectively. We then define

$$\phi(\boldsymbol{U}^R, \boldsymbol{U}^I, \boldsymbol{\Lambda}) = \left[ \phi(\boldsymbol{U}_1^R \,\|\, \boldsymbol{U}_1^I \,\|\, \boldsymbol{\Lambda} + \epsilon), \ldots, \phi(\boldsymbol{U}_n^R \,\|\, \boldsymbol{U}_n^I \,\|\, \boldsymbol{\Lambda} + \epsilon) \right] \in \mathbb{R}^{n \times n \times d},$$

where $\phi \colon \mathbb{R}^3 \to \mathbb{R}^d$ is applied row-wise, on the last dimension of $\boldsymbol{U}_1^R \,\|\, \boldsymbol{U}_1^I \,\|\, \boldsymbol{\Lambda} + \epsilon \in \mathbb{R}^{n \times n \times 3}$ and $\epsilon \in \mathbb{R}^l$ is a learnable, zero-initialized vector. The mLPE is then obtained by aggregating across eigenvectors as

$$\text{mLPE}(\boldsymbol{U}^R, \boldsymbol{U}^I, \boldsymbol{\Lambda}) = \rho\left( \sum_{i=1}^n \phi(\boldsymbol{U}^R, \boldsymbol{U}^I, \boldsymbol{\Lambda})_i^\top \right),$$

with $\rho \colon \mathbb{R}^{n \times d} \to \mathbb{R}^d$ is an equivariant feed-forward neural networks.

To characterize adjacency recovery in the directed setting, we introduce the notion of *out-adjacency-identifiability*. Given node embeddings $\boldsymbol{P} \in \mathbb{R}^{n \times d}$, we compute

$$\tilde{\boldsymbol{P}}^R = \tfrac{1}{\sqrt{d}}(\boldsymbol{P}\boldsymbol{W}^{K_R})(\boldsymbol{P}\boldsymbol{W}^{Q_R})^\top, \qquad \tilde{\boldsymbol{P}}^I = \tfrac{1}{\sqrt{d}}(\boldsymbol{P}\boldsymbol{W}^{K_I})(\boldsymbol{P}\boldsymbol{W}^{Q_I})^\top,$$

with learnable matrices $\boldsymbol{W}^{K_R}, \boldsymbol{W}^{Q_R}, \boldsymbol{W}^{K_I}, \boldsymbol{W}^{Q_I} \in \mathbb{R}^{d \times d}$, and combine them as

$$\tilde{\boldsymbol{P}} := \tilde{\boldsymbol{P}}^R + \tfrac{2-c}{s}\, \tilde{\boldsymbol{P}}^I, \quad c = \cos(2\pi q), \quad s = \sin(2\pi q).$$

We say that $\boldsymbol{P}$ is out-adjacency-identifying if for each node $i$,

$$\tilde{\boldsymbol{P}}_{ij} = \max_k \tilde{\boldsymbol{P}}_{ik} \iff \boldsymbol{A}_{ij} = 1,$$

We further say that a matrix $\boldsymbol{Q} \in \mathbb{R}^{n \times d}$ is *sufficiently out-adjacency-identifying* if it approximates an adjacency-identifying matrix $\boldsymbol{P}$ to arbitrary precision, i.e., $\|\boldsymbol{P} - \boldsymbol{Q}\|_F < \epsilon$, for all $\epsilon > 0$.

Now, the result below shows that the mLPE produces node embeddings from which the full directed adjacency of a DAG can be recovered with arbitrarily high accuracy. In other words, no structural information is lost in the encoding step.

**Theorem 1.** *The magnetic Laplacian positional encoding (mLPE) is sufficiently out-of-adjacency-identifying.*

Proofs and the full DAG autoencoder specification are provided in Appendix C.2. This result ensures that, by replacing the LPE with the mLPE, the LG-VAE naturally extends to DAGs while preserving node-level latent representations that are compatible with latent diffusion.

## 5 EXPERIMENTAL STUDY

We now investigate the performance of our approach in relation to the requirements we derived in the introduction. Specifically, we answer the following questions.

**Q1** How faithfully does LG-VAE reconstruct the graph's structure?

**Q2** How does LG-Flow compare to prior graph diffusion models in generative performance?

**Q3** How efficient is LG-Flow in inference time compared to state-of-the-art graph diffusion models?

**Datasets** We evaluate the generative performance of our approach on six datasets: three synthetic datasets–EXTENDED PLANAR, EXTENDED TREE, and EGO–, two molecular datasets–MOSES and GUACAMOL–, and TPU TILE, a DAG generation datasets. An extensive description of those datasets is available in Appendix E.1.

### 5.1 Q1: AUTOENCODER RECONSTRUCTION ABILITY

We begin by empirically evaluating our autoencoder's ability to reconstruct adjacency matrices faithfully. We focus on two simple, unlabeled datasets with strong structural constraints—EXTENDED PLANAR and EXTENDED TREE. For these distributions, near-lossless reconstruction is crucial to ensure that decoded graphs maintain their structural properties, specifically planarity and tree structure.

**Baselines** We evaluate LG-VAE on EXTENDED PLANAR and TREE, and compare it against two prior graph autoencoders: DGVAE (Boget et al., 2024) and the autoencoder used in GLAD (Nguyen et al., 2024). We also conduct ablations of LG-VAE's components, (i) removing the positional encoder $\phi$ to obtain a raw GNN ENCODER, and (ii) replacing our decoder with the original GVAE DECODER, yielding a GVAE-like variant that retains $\phi$ as positional encoder.

**Metrics** Our primary evaluation metric is SAMPLE ACCURACY, i.e., the fraction of test graphs that the

Table 2: **Autoencoder reconstruction performance.** Best results are in **bold**. Only LG-VAE achieves near-lossless reconstruction, as shown by the SAMPLE ACCURACY metric.

| | Method | EXTENDED PLANAR | | EXTENDED TREE | |
|---|---|---|---|---|---|
| | | Edge Acc. | Sample Acc. | Edge Acc. | Sample Acc. |
| | GLAD | 0.9129 | 0. | 0.9866 | 0. |
| | DGVAE | 0.9547 | 0. | 0.9777 | 0. |
| Ours | GVAE DECODER | 0.9000 | 0. | 0.6351 | 0. |
| | GNN ENCODER | 0.9977 | 0.7734 | 0.2539 | 0. |
| | LG-VAE | **1.0000** | **0.9961** | **1.0000** | **0.9844** |

decoder reconstructs exactly, including node/edge labels when present. We also report EDGE ACCURACY, i.e., the proportion of correctly reconstructed edges in the test set.

**Results** We report results in Table 2. Except for LG-VAE, no method achieves near-lossless reconstruction in terms of SAMPLE ACCURACY. However, all methods attain EDGE ACCURACY above 0.9, only LG-VAE and its raw GNN ENCODER variant obtain non-zero SAMPLE ACCURACY.

Consequently, these findings highlight the limitations of current VAEs, whose latent representations are less expressive than those of LG-VAE. They struggle to reconstruct samples from latents with high accuracy, casting doubt on their ability to recover the required structural properties. For instance, even if a latent diffusion model perfectly matches the latent distribution, the decoder is not guaranteed to yield planar graphs or trees.

## 5.2 **Q2 & Q3:** ASSESSING GENERATIVE PERFORMANCE AND INFERENCE EFFICIENCY

In this section, we evaluate our latent diffusion model on various benchmarks, including synthetic graphs, molecules, and DAGs.

**Baselines** We evaluate LG-Flow against several prior graph diffusion models, two discrete-time methods, DIGRESS (Vignac et al., 2023) and its sparse counterpart, SPARSEDIFF (Qin et al., 2023), two continuous-time methods, COMETH (Siraudin et al., 2024) and DEFOG (Qin et al., 2025). Note that DISCO (Xu et al., 2024) and COMETH are highly similar methods, and either could have been evaluated. We chose to focus on COMETH due to the inherent high computational cost of discrete graph diffusion models. On DAGs, we assess our approach against diffusion models tailored to DAGs, namely LAYERDAG, an autoregressive diffusion model that generates DAGs layer by layer, and DIRECTO, a recent adaptation of DEFOG to DAG generation.

**Metrics** For synthetic graphs and DAGs, we evaluate generation quality using the standard *Maximum Mean Discrepancy* (MMD) metrics between test and generated sets, computed on various statistics, e.g., degree or clustering coefficient. Validity for trees, planar graphs, and DAGs is defined by structural constraints (planarity, tree property, or acyclicity). For molecular datasets, we assess validity based on fundamental chemical rules, uniqueness, and novelty, complemented by benchmark-specific scores. A detailed description of all metrics is deferred to Appendix E.1.

Across all datasets, we also measure sampling time (**Time**) to assess inference efficiency. All models were sampled using equal batch sizes whenever possible, or otherwise with batch sizes chosen to maximize GPU utilization.

**Results on synthetic graphs** Our results are shown in Table 3a. Our approach is competitive across all distribution metrics and achieves strong performance on V.U.N. In terms of sampling time, it delivers a $50\times$ speed-up on EXTENDED PLANAR and a $10\times$ speed-up on EXTENDED TREE. On EGO, our method matches the performance of DEFOG while reducing sampling time by up to $1000\times$.

To further demonstrate the efficiency of LG-Flow, we plot memory requirements during sampling as a function of batch size in Table 3. Due to the DiT's efficient implementation, the memory usage of our method scales linearly with the batch size, allowing the entire test batch of 151 samples to fit into memory. In contrast, DEFOG runs out of memory once the batch size exceeds 32.

**Results on molecular generation** Our results are summarized in Table 4. On GUACAMOL, our method achieves strong performance on validity, uniqueness, and novelty, consistently outperforming DIGRESS and DISCO. On **KL div**, we reach state-of-the-art results, with performance close to DEFOG. Most notably, we surpass all baselines on **FCD** by a large margin, highlighting the method's ability to capture the underlying chemical distribution.

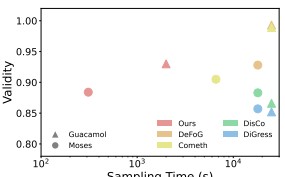

Figure 3: We compare molecular validity and sampling time. The ideal lies in the top-left corner, with high validity and low cost. Our method offers a better trade-off than prior graph-space diffusion models.

On MOSES, LG-Flow remains competitive across all metrics, outperforming DIGRESS and DISCO on validity again. Crucially, while maintaining competitive accuracy, our method delivers substantial inference efficiency gains, achieving a $36.4\times$ speedup on GUACAMOL and a $58\times$ speedup on MOSES compared to the state-of-the-art DEFOG. To put

Table 3: **Generation results on synthetic graphs.** In the table, best results are in **bold**, second best are underlined. In Table 3, empty squares indicate out-of-memory errors. While our method scales linearly with batch size and can accommodate the entire test batch of 151 samples on a single GPU, DEFOG runs out of memory at batch size 64. MMD metrics are reported with a $10^3$ factor.

(a) Generation results on the Extended Planar and Extended Tree datasets.

| Dataset | Extended Planar | | | | | | Extended Tree | | | | | |
|---|---|---|---|---|---|---|---|---|---|---|---|---|
| Method | Deg. (↓) | Orbit (↓) | Cluster. (↓) | Spec. (↓) | V.U.N. (↑) | Time (↓) | Deg. (↓) | Orbit (↓) | Cluster. (↓) | Spec. (↓) | V.U.N. (↑) | Time (↓) |
| SparseDiff | 0.8±0.1 | 0.9±0.1 | 20.9±1.3 | 1.9±0.2 | 74.1±2.1 | $12.1 \times 10^2$ | **0.**±0. | 0.±0. | 0.±0. | **1.2**±0.2 | 95.3±1.5 | $11.5 \times 10^2$ |
| Cometh | 0.7±0.1 | 2.2±0.3 | 20.0±1.1 | 1.3±0.2 | 96.7±0.8 | $9.7 \times 10^2$ | **0.**±0. | 0.±0. | 0.±0. | 1.4±0.4 | 93.8±1.6 | $9.7 \times 10^2$ |
| DeFog | **0.3**±0.1 | **0.1**±0.1 | **9.7**±0.3 | 1.3±0.1 | **99.6**±0.4 | $8.9 \times 10^2$ | 0.1±0. | 0.±0. | 0.±0. | **1.2**±0.2 | **98.1**±0.7 | $8.9 \times 10^2$ |
| LG-Flow | **0.3**±0.1 | 0.5±0.2 | 10.1±3.1 | **1.0**±0.2 | 99.0±0.2 | **4.6**±0. | 0.1±0. | 0.±0. | 0.±0. | 1.6±0.3 | 93.3±0.8 | **9.0**±0. |

(b) Generation results on large graphs: Ego.

| Dataset | Ego | | | | | |
|---|---|---|---|---|---|---|
| Method | Deg. (↓) | Orbit (↓) | Cluster. (↓) | Spec. (↓) | Ratio (↓) | Time (s) (↓) |
| Training Set | 0.2 | 6.8 | 7.1 | 1.0 | 1.0 | – |
| DiGress | 8.9±1.6 | 30±3 | 54±4 | 19±3.2 | 19±3.1 | – |
| SparseDiff | 3.7±0.4 | 20±4 | 32±1 | **5.6**±0.8 | 7.9±0.9 | $5.3 \times 10^2$ |
| Cometh | 8.1±0.1 | 19.6±2.0 | 37.4±1.5 | 12.9±1.6 | 15.4±1.7 | $1.3 \times 10^4$ |
| DeFog | **1.8**±0.5 | 21.4±2.6 | 30.7±3.3 | 6.2±1.2 | **5.2**±0.8 | $1.3 \times 10^4$ |
| LG-Flow | 1.9±0.5 | **15.5**±3.8 | 22.4±2.8 | 6.2±1.1 | 5.3±1.0 | **11.6**±0.6 |

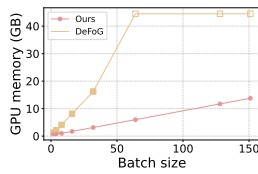

(c) Comparison of memory use for DE-FOG and our method on the Ego dataset.

Table 4: **Generation on molecular datasets : MOSES & GuacaMol**. Best results are denoted in **bold** and second best are underlined.

| Dataset | GuacaMol | | | | | | MOSES | | | | | | | |
|---|---|---|---|---|---|---|---|---|---|---|---|---|---|---|
| Model | V. (↑) | V.U. (↑) | V.U.N (↑) | KL div (↑) | FCD (↑) | Time (s) (↓) | Val. (↑) | Unique. (↑) | Novel. (↑) | Filters (↑) | FCD (↓) | SNN (↑) | Scaf. (↑) | Time (s) (↓) |
| Training Set | 100.0 | 100.0 | 0.0 | 99.9 | 92.8 | – | 100.0 | 100.0 | 0.0 | 100.0 | 0.01 | 0.64 | 99.1 | – |
| DiGress | 85.2 | 85.2 | 85.1 | 92.9 | 68.0 | – | 85.7 | **100.0** | 95.0 | 97.1 | 1.19 | 0.52 | 14.8 | – |
| DisCo | 86.6 | 86.6 | 86.5 | 92.6 | 59.7 | – | 88.3 | **100.0** | 97.7 | 95.6 | 1.44 | 0.50 | 15.1 | – |
| Cometh | 98.9 | 98.9 | 97.6 | 92.6 | 72.7 | $4.0 \times 10^4$ | 90.5 | 99.9 | 92.6 | **99.1** | 1.27 | 0.54 | **16.0** | $1.9 \times 10^4$ |
| DeFoG | **99.0** | **99.0** | **97.9** | 97.7 | 73.8 | $4.0 \times 10^4$ | 92.8 | 99.9 | 92.1 | 98.9 | 1.95 | **0.55** | 14.4 | $1.8 \times 10^4$ |
| LG-Flow | 93.0 | 93.0 | 91.5 | 97.2 | **81.8** | $1.1 \times 10^3$ | 88.4 | 99.9 | 90.5 | 98.9 | 1.43 | **0.55** | 12.3 | $3.1 \times 10^2$ |

these results into perspective, we plot validity against sampling efficiency in Figure 3. Our method exhibits a significantly better trade-off than prior diffusion models that operate directly in graph space.

**Results on DAG generation** Our results are reported in Table 5. As already noted in Carballo-Castro et al. (2025), LAYERDAG's strong performance on **V.U.N** is misleading: it collapses on distributional metrics, clearly failing to capture the underlying distributional characteristics.

In contrast, we outperform DIRECTO on **Cluster.**, **Spec.**, **Wave.**, and, most importantly, on **Valid**. LG-Flow shows slightly higher memorization than DIRECTO, likely due to the limited diversity of the training data. Most importantly, our approach is dramatically more efficient at inference, achieving up to a $680\times$ speed-up over DIRECTO.

Table 5: **Generation on DAGs: TPU Tile dataset.** We highlight best results in **bold**.

| Method | Out Deg. | In Deg. | Cluster. | Spec. | Wave. | Valid | Unique | Novel | V.U.N | Time |
|---|---|---|---|---|---|---|---|---|---|---|
| Training Set | 0.3 | 0.3 | 0.7 | 0.6 | 0.2 | 1.0 | 1.0 | 0. | 0. | – |
| LayerDAG | 193.3±90.5 | 222.5±39.5 | 151.2±52.2 | 50.1±20.6 | 76.5±25.1 | **100.0**±0.0 | **99.5**±1.0 | **98.5**±3.0 | **98.5**±3.0 | – |
| Directo | **3.9**±1.7 | **37.6**±5.1 | 21.1±11.7 | 12.6±2.2 | 6.2±0.9 | 90.5±3.3 | 90.5±4.6 | 97.5±3.2 | 80.5±4.6 | $3.4 \times 10^2$ |
| LG-Flow | 12.8±3.6 | 81.8±6.3 | **5.9**±2.5 | **8.1**±1.3 | **4.0**±1.3 | 98.5±2.6 | 94.0±3.3 | 77.5±3.9 | 74.5±3.8 | **0.5**±0.3 |

# 6 CONCLUSION

We proposed a latent diffusion framework for graphs, built on a permutation-equivariant Laplacian autoencoder with provable reconstruction guarantees for undirected graphs and DAGs. By mapping nodes to fixed-dimensional embeddings, our method removes the quadratic cost of graph-space diffusion and supports high-capacity backbones such as Diffusion Transformers. Experiments show near-lossless reconstruction, competitive or superior generative performance, and sampling up to three orders of magnitude faster. This framework shifts the focus from graph-specific diffusion design to cheaper, more expressive autoencoders, unifies graphs and DAGs generation, enables transferring conditional methods from image generation, and paves the way for scaling generative models to much larger graphs.

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

# Appendices

## A    EXTENDED RELATED WORK

Here, we discuss more related work.

**Graph generation**    Graph generation methods are typically categorized into two main groups. *Autoregressive models* build graphs by progressively adding nodes and edges (You et al., 2018; Jang et al., 2024). Their main advantage is computational efficiency, since they do not need to materialize the full adjacency matrix. However, they depend on an arbitrary node ordering to transform the graph into a sequence, either learned or defined through complex algorithms, which breaks permutation equivariance. In contrast, *one-shot models* generate the entire graph at once, thereby preserving permutation equivariance. Many approaches have been explored in this class, starting with GVAEs (Kipf and Welling, 2016) and GANs (Martinkus et al., 2022). Following their success in other modalities such as images, diffusion models (Ho et al., 2020; Lipman et al., 2023; Song et al., 2021) quickly emerged as the dominant approach for graph generation.

**Graph diffusion models**    Diffusion models can be grouped according to two criteria: whether they operate in continuous or discrete state spaces, and whether they train on a continuous or discrete time axis. Discrete-time diffusion models (Austin et al., 2021; Ho et al., 2020) were successfully adapted to graph generation, with discrete-state models (Chen et al., 2023b; Vignac et al., 2023) showing an advantage over their continuous-state counterparts (Jo et al., 2022; Niu et al., 2020). However, these models relied on a fixed time discretization during training, which limited their flexibility. They were later extended to continuous-time, both in continuous (Lipman et al., 2023; Song et al., 2021) and discrete state spaces (Campbell et al., 2022; Gat et al., 2024; Lou et al., 2024), and also adapted to graph generation, once again confirming the superiority of discrete-state approaches (Siraudin et al., 2024; Qin et al., 2025; Xu et al., 2024). Previous graph diffusion models relied on Laplacian decompositions but pursue objectives that differ from ours. Regarding Minello et al. (2025), the approach is fundamentally different from ours: it generates approximate eigenvalues and eigenvectors and then reconstructs the adjacency matrix through a separate prediction module, whereas we embed structural information in the latent space by feeding eigenvectors and eigenvalues to the encoder. Concerning Zhu et al. (2025), the scope of the paper is different from ours. The authors focus on representation learning using graph diffusion models, while our method is evaluated solely for generation. In contrast to traditional graph diffusion models that target the whole adjacency matrix, they train their model to reconstruct only the low-frequency Laplacian eigenvectors. Overall, these methods aim to reconstruct spectral components of the Laplacian, whereas we use them to encode structural information into the latent space.

**Latent diffusion models**    *Latent diffusion models* (LDMs) are a popular framework, first introduced for image generation (Rombach et al., 2022). They extend traditional diffusion models by operating in a compressed latent space rather than directly in pixel space. An autoencoder—typically a VAE—is used to encode high-resolution data into a lower-dimensional latent representation. The diffusion process then unfolds within this latent space, substantially reducing computational cost, while the decoder reconstructs fine-grained details. This paradigm underlies widely used models such as Stable Diffusion. More recently, DiTs (Peebles and Xie, 2023) have been introduced as the main denoising architecture and have achieved strong results across modalities (Esser et al., 2024; Joshi et al., 2025). Although some works have attempted to adapt latent diffusion to graph generation, none have demonstrated competitive performance compared to prior state-of-the-art discrete diffusion models (Fu et al., 2024; Nguyen et al., 2024; Yang et al., 2024a). To the best of our knowledge, only Nguyen et al. (2024) evaluate the reconstruction performance of their autoencoder, and only on the QM9 dataset, which contains very small graphs. Additionally, they employ an ad-hoc transformer architecture that materializes the full attention score matrix, which limits computational efficiency. While Zhou et al. (2024) appears to report competitive results on MOSES, it encodes the full adjacency matrix through $n^2$ edge tokens, with $n$ the number of nodes, which prevents efficiency gains. Other works are specific to molecule generation (Bian et al., 2024; Ketata et al., 2025; Wang et al., 2024). Finally, some focus on tasks other than graph generation, such as link prediction (Fu et al., 2024) or regression (Zhou et al., 2024). The key difference between these works and ours is that we design our autoencoder to ensure that the adjacency matrix can be provably recovered from the latent space.

## B    ADDITIONAL BACKGROUND AND NOTATIONS

Here, we provide additional notation and background.

### B.1    NOTATIONS

Let $\mathbb{N} := \{1, 2, \dots\}$ and for $n \in \mathbb{N}$ define $[n] := \{1, \dots, n\} \subset \mathbb{N}$. We denote by $\mathbb{R}^{+*}$ the set of strictly positive real numbers. We consider *node- and edge-labeled undirected graphs* $G$, i.e., tuples $(V(G), E(G), \ell_x, \ell_e)$ with up to $n$ nodes and $m$ edges, without self-loops or isolated nodes. Here, $V(G)$ is the set of nodes, $E(G)$ the set of edges, $\ell_x : V(G) \to [a]$, for $a \in \mathbb{N}$, assigns each node one of $a$ discrete labels, and $\ell_e : E(G) \to [b]$, for $b \in \mathbb{N}$, assigns each edge one of $b$ discrete labels. An undirected edge $\{u, v\} \in E(G)$ is written either $(u, v)$ or $(v, u)$. We also consider *node- and edge-labeled directed acyclic graphs* (DAGs), defined analogously except that $(u, v)$ denotes a directed edge from $u$ to $v$ and the graph contains no directed cycles. Throughout, we assume an arbitrary but fixed ordering of nodes so that each node corresponds to a number in $[n]$. The adjacency matrix of $G$ is denoted $\boldsymbol{A}(G) \in \{0, 1\}^{n \times n}$, where $\boldsymbol{A}(G)_{ij} = 1$ if and only if nodes $i$ and $j$ are connected. We construct a node feature vector $\boldsymbol{X} \in [a]^n$ consistent with $\ell_x$, so that $X_i = X_j$ if and only if $\ell_x(i) = \ell_x(j)$. Similarly, we define an edge feature matrix $\boldsymbol{E} \in [b]^m$ consistent with $\boldsymbol{A}$ and $\ell_e$, so that $E_{ij} = E_{kl}$ if and only if $\ell_e((i, j)) = \ell_e((k, l))$. For any matrix $\boldsymbol{M}$, we denote by $\boldsymbol{M}_i$ its $i$-th row. For a real-valued matrix $\boldsymbol{M} \in \mathbb{R}^{n \times n}$, $\boldsymbol{M}^\top$ denotes its transpose, and for a complex matrix $\boldsymbol{N}$, $\boldsymbol{N}^*$ denotes its Hermitian transpose. For two binary matrices $\boldsymbol{P}$ and $\boldsymbol{Q}$, we denote their logical disjunction $\boldsymbol{P} \vee \boldsymbol{Q}$ where $(\boldsymbol{P} \vee \boldsymbol{Q})_{ij} = 1$ if $\boldsymbol{P}_{ij} = 1$ or $\boldsymbol{Q}_{ij} = 1$, and 0 otherwise. Let $\boldsymbol{u}$ and $\boldsymbol{v}$ be two vectors; $\boldsymbol{u} \odot \boldsymbol{v}$ denotes the element-wise product between them. Finally, the *graph Laplacian* is $\boldsymbol{L} := \boldsymbol{D} - \boldsymbol{A}$, where $\boldsymbol{D} \in \mathbb{N}^{n \times n}$ is the degree matrix. Its eigenvalue decomposition is $\boldsymbol{L} := \boldsymbol{U} \boldsymbol{\Lambda} \boldsymbol{U}^\top$, where $\boldsymbol{\Lambda} := (\lambda_1, \dots, \lambda_n)$ is the vector of eigenvalues (possibly repeated) and $\boldsymbol{U} \in \mathbb{R}^{n \times n}$ the matrix of eigenvectors, with the $i$-th column $\boldsymbol{U}_i^\top$ corresponding to eigenvalue $\lambda_i$. We denote by $\boldsymbol{I}_d$ the $d$-dimensional identity matrix.

**Flow Matching**    *Flow Matching* (FM) (Lipman et al., 2023) is a generative modeling framework that transports samples from a base distribution $p_0$ to a target distribution $p_1$ by gradually denoising them. The method relies on a time-dependent velocity field $v_t : [0, 1] \times \mathbb{R}^d \to \mathbb{R}^d$, which defines a *flow map* $\psi_t : [0, 1] \times \mathbb{R}^d \to \mathbb{R}^d$ through the ordinary differential equation

$$\frac{d}{dt} \psi_t(\boldsymbol{x}) = v_t\big(\psi_t(\boldsymbol{x})\big), \qquad \psi_0(\boldsymbol{x}) := \boldsymbol{x}.$$

The flow induces a family of intermediate distributions $(p_t)_{t \in [0,1]}$, where $p_t$ is the *pushforward* of $p_0$ by $\psi_t$. In other words, if $\boldsymbol{x}_0 \sim p_0$, then $\psi_t(\boldsymbol{x}_0) \sim p_t$, and in particular $\psi_1(\boldsymbol{x}_0) \sim p_1$. The *probability path* $(p_t)$ is arbitrary as long as it is differentiable in time and satisfies the boundary conditions at $t = 0$ and $t = 1$.

In practice, the true velocity field $v_t$ is intractable. FM therefore relies on conditional probability paths $p_t(\boldsymbol{x} \mid \boldsymbol{x}_1)$ for which the conditional velocity $u_t(\boldsymbol{x} \mid \boldsymbol{x}_1)$ can be computed in closed form. A standard choice is the linear interpolation

$$\boldsymbol{x}(t) := (1 - t)\boldsymbol{x}_0 + t\boldsymbol{x}_1, \qquad \boldsymbol{x}_0 \sim p_0, \ \boldsymbol{x}_1 \sim p_1,$$

with corresponding velocity

$$u_t(\boldsymbol{x} \mid \boldsymbol{x}_1) := \frac{\boldsymbol{x}_1 - \boldsymbol{x}}{1 - t}.$$

To train a generative model, the intractable $v_t$ is replaced by a neural network $v_\theta(t, \boldsymbol{x})$, with parameters $\theta$ that learns to match the conditional velocities $u_t$. This is achieved by minimizing the *conditional flow-matching loss*

$$\mathcal{L}(\theta) := \mathbb{E}_{t, \boldsymbol{x}_0, \boldsymbol{x}_1} \left[ \big\| v_\theta\big(t, (1 - t)\boldsymbol{x}_0 + t\boldsymbol{x}_1\big) - (\boldsymbol{x}_1 - \boldsymbol{x}_0) \big\|^2 \right].$$

This formulation avoids integrating the ODE during training, which makes FM both efficient and straightforward to implement. After training, new samples are obtained by combining the learned velocity field $v_\theta$ from $t = 0$ to $t = 1$, yielding $\boldsymbol{x}_1 \sim p_1$ deterministically. We refer to Lipman et al. (2024) for a comprehensive description of the framework. Since flow matching and Gaussian diffusion are essentially equivalent, we use the two terms interchangeably (Gao et al., 2024).

**Variational Autoencoders**   A *Variational Autoencoder* (VAE), also referred to as a KL-regularized autoencoder, is a latent-variable generative model that extends the autoencoding framework with a probabilistic formulation. Given data $\boldsymbol{x} \in \mathbb{R}^{n \times d}$, the encoder parameterizes an approximate posterior $q_\phi(\boldsymbol{z} \mid \boldsymbol{x})$, where $\boldsymbol{z}$ denotes the latent, $\boldsymbol{z} \in \mathbb{R}^{n \times p}, p \ll d$. The decoder defines the conditional likelihood $p_\theta(\boldsymbol{x} \mid \boldsymbol{z})$. Training maximizes the *Evidence Lower Bound* (ELBO),

$$\mathcal{L}(\theta, \phi; \boldsymbol{x}) := \mathbb{E}_{q_\phi(\boldsymbol{z}|\boldsymbol{x})}[\log p_\theta(\boldsymbol{x} \mid \boldsymbol{z})] - D_{\mathrm{KL}}(q_\phi(\boldsymbol{z} \mid \boldsymbol{x}) \,\|\, p(\boldsymbol{z})),$$

where the first term enforces reconstruction fidelity and the second regularizes the posterior toward a prior $p(\boldsymbol{z})$, typically $\mathcal{N}(0, \boldsymbol{I}_p)$. To enable backpropagation through the latents, the reparameterization trick is employed, i.e, the encoder predicts the mean and log-variance of the posterior distribution, $\boldsymbol{\mu}_\phi$ and $\log \boldsymbol{\sigma}_\phi$, and $\boldsymbol{z}$ is sampled according to

$$\boldsymbol{z} = \boldsymbol{\mu}_\phi(\boldsymbol{x}) + \boldsymbol{\sigma}_\phi(\boldsymbol{x}) \, \epsilon, \quad \epsilon \sim \mathcal{N}(0, \boldsymbol{I}_p).$$

This formulation yields a continuous and structured latent space, supporting interpolation and generative sampling.

**GIN**   A *Graph Isomorphism Network* (GIN) is a message-passing graph neural network designed to maximize expressive power under the neighborhood aggregation framework (Xu et al., 2019). Let each node $i$ have features $\boldsymbol{h}_i^{(l)}$ at layer $l > 0$, and let $N(i)$ denote its neighbors. GIN updates node representations via

$$\boldsymbol{h}_i^{(l+1)} := \mathrm{MLP}^{(l+1)}\Big((1 + \varepsilon)\, \boldsymbol{h}_i^{(l)} + \sum_{j \in N(i)} \boldsymbol{h}_j^{(l)}\Big),$$

and $\boldsymbol{h}_i^{(0)}$ is the initial node feature after being fed through an MLP. A natural extension, termed GINE, incorporates edge features into this aggregation by modifying each neighbor's contribution, formally,

$$\boldsymbol{h}_i^{(l+1)} := \mathrm{MLP}^{(l+1)}\Big((1 + \varepsilon)\, \boldsymbol{h}_i^{(l)} + \sum_{j \in N(i)} (\boldsymbol{h}_j^{(l)} + \boldsymbol{e}_{j,i}^{(l)})\Big),$$

where $\boldsymbol{e}_{j,i}^{(l)}$ denotes the edge feature between nodes $j$ and $i$; this model retains the aggregation mechanism of GIN while leveraging edge-level information.

**Diffusion Transformers (DiT)**   A *Diffusion Transformer* (DiT) is a diffusion-based generative model in which the denoising backbone is a Transformer model. Importantly, conditioning signals—such as the timestep $t$ or additional context $c$—are incorporated via adaptive layer normalization (AdaLN) within each block, following a formulation of the form,

$$\mathrm{AdaLN}(\boldsymbol{h}, c) := \gamma(c)\, \mathrm{LayerNorm}(\boldsymbol{h}) + \beta(c),$$

where $\gamma(c)$ and $\beta(c)$ are learned affine modulations derived from $c$. This injection method preserves the scalability and expressive capacity of standard Transformers while enabling effective conditional generation across data modalities.

## C   METHOD DETAILS AND PROOFS

Here, we describe additional details and outline missing proof from the main paper.

### C.1   DECODING EDGE LABELS

As a practical solution to decode edge labels along with the graph structure, we use a multi-headed version of the bilinear layer, where $(\boldsymbol{W}_h^Q, \boldsymbol{W}_h^K) \in \mathbb{R}^{d \times d}, h \in [b + 1]$ define $b$ parallel projection heads, corresponding to the $b$ possible edge labels, and an additional head for the absence of an edge. Formally,

$$\tilde{\boldsymbol{A}}_h := \frac{1}{\sqrt{d}} \boldsymbol{Z} \boldsymbol{W}_h^Q (\boldsymbol{Z} \boldsymbol{W}_h^K)^\top \in \mathbb{R}^{n \times n}, \quad h \in [b + 1],$$

$$\tilde{\boldsymbol{A}} := \Big[\mathrm{DeepSet}\left(\tilde{\boldsymbol{A}}_0\right), \dots, \mathrm{DeepSet}\left(\tilde{\boldsymbol{A}}_b\right)\Big] \in \mathbb{R}^{n \times n \times (b+1)},$$

$$\hat{\boldsymbol{A}} := \mathcal{D}_e(\boldsymbol{Z}) = \mathrm{softmax}(\tilde{\boldsymbol{A}}) \in \mathbb{R}^{n \times n \times (b+1)},$$

where the $\mathrm{softmax}$ is applied over the last dimension of $\hat{\boldsymbol{A}}$. At inference, the node (respectively edge) labels are given by the argmax over the scores across the $a$ (resp. $b + 1$) classes.

## C.2 Formal definition of DAG auto-encoder

In this section, we formally define the architecture of our auto-encoder for DAGs.

**Encoder** We instantiate $\rho$, as GIN architecture, wrapped into a PyTorch Geometric's DirGNNConv (Rossi et al., 2023), which use one layer for in-neighbors and one for out-neighbors. We implement $\phi$ as a row-wise MLP.

The encoder is defined in a similar way to that for undirected graphs, except it takes both the real and imaginary parts of the eigenvectors as input. Given a DAG $G$ with node features $\boldsymbol{X} \in \mathbb{R}^n$, edge features $\boldsymbol{E} \in \mathbb{R}^{n \times n}$, and magnetic Laplacian $\boldsymbol{L}_q = \boldsymbol{U} \boldsymbol{\Lambda} \boldsymbol{U}^*$, the encoder produces the mean $\boldsymbol{\mu_Z} \in \mathbb{R}^d$ and log-variance $\log \boldsymbol{\sigma_Z}^2 \in \mathbb{R}^d$ of a Gaussian variational posterior:

$$\boldsymbol{H}_\phi := \sum_{i=1}^n \phi(\boldsymbol{U}^R, \boldsymbol{U}^I, \boldsymbol{\Lambda})_i^\top \in \mathbb{R}^{n \times d},$$

$$\boldsymbol{H}_X := \boldsymbol{X} \boldsymbol{W}^X + \boldsymbol{b}^X \in \mathbb{R}^{n \times d},$$

$$\boldsymbol{H}_E := \boldsymbol{E} \boldsymbol{W}^E + \boldsymbol{b}^E \in \mathbb{R}^{m \times d},$$

$$\boldsymbol{\mu_Z}, \log \boldsymbol{\sigma_Z}^2 = \mathcal{E}(G, \boldsymbol{X}, \boldsymbol{E}, \boldsymbol{U}^R, \boldsymbol{U}^I, \boldsymbol{\Lambda}) = \rho\left(\boldsymbol{H}_X, \boldsymbol{H}_\phi, \boldsymbol{H}_E\right).$$

A latent sample $\boldsymbol{Z} \in \mathbb{R}^{n \times d}$ is then obtained using the reparameterization trick,

$$\boldsymbol{Z} = \boldsymbol{\mu_Z} + \boldsymbol{\sigma_Z} \odot \epsilon, \quad \epsilon \sim \mathcal{N}(0, \boldsymbol{I}_d),$$

where $\boldsymbol{\sigma_Z} = \exp\left(\frac{1}{2} \log \boldsymbol{\sigma_Z}^2\right)$.

**Decoder** Similarly to the undirected case, the node decoder $\mathcal{D}_x$ reconstructs node labels directly from the latents $\boldsymbol{Z}$. Then an edge decoder $\mathcal{D}_e$ is defined consistently with our definition of out-adjacency-identifying encodings, as follows,

$$\hat{\boldsymbol{X}} := \mathcal{D}_x(\boldsymbol{Z}) = \boldsymbol{Z} \boldsymbol{W}^{D_x} + \boldsymbol{b}^{D_x}, \quad \hat{\boldsymbol{X}} \in \mathbb{R}^{n \times a},$$

$$\tilde{\boldsymbol{Z}}_R := \frac{1}{\sqrt{d}} \boldsymbol{Z} \boldsymbol{W}_R^Q (\boldsymbol{Z} \boldsymbol{W}_R^K)^\top$$

$$\tilde{\boldsymbol{Z}}_I := \frac{1}{\sqrt{d}} \boldsymbol{Z} \boldsymbol{W}_I^Q (\boldsymbol{Z} \boldsymbol{W}_I^K)^\top$$

$$\tilde{\boldsymbol{Z}} := \tilde{\boldsymbol{Z}}_R + \frac{2-c}{s} \tilde{\boldsymbol{Z}}_I$$

$$\hat{\boldsymbol{A}} := \mathcal{D}_e(\boldsymbol{Z}) = \sigma\left(\text{DeepSet}\left(\tilde{\boldsymbol{Z}}\right)\right), \quad \hat{\boldsymbol{A}} \in \mathbb{R}^{n \times n}$$

where $c := \cos(2\pi q)$ and $s := \sin(2\pi q)$, $\boldsymbol{W}^X \in \mathbb{R}^{d \times a}$ and $\boldsymbol{W}_R^Q, \boldsymbol{W}_R^K, \boldsymbol{W}_I^Q, \boldsymbol{W}_I^K \in \mathbb{R}^{d \times d}$ are a learnable weight matrices and the MLP is applied row-wise.

## C.3 Proofs

We first prove the following lemma, which provides a sufficient condition for node embeddings to be out-adjacency-identifying.

**Lemma 2.** *Let $G$ be a directed graph with magnetic Laplacian $\boldsymbol{L}_q$. Let $\boldsymbol{P} \in \mathbb{R}^{n \times d}$ a matrix of node embeddings . If there exists $\boldsymbol{W}^{K_R}, \boldsymbol{W}^{Q_R}, \boldsymbol{W}^{K_I}, \boldsymbol{W}^{Q_I} \in \mathbb{R}^{d \times d}$ such that*

$$\tilde{\boldsymbol{P}}^R = \frac{1}{\sqrt{d}} (\boldsymbol{P} \boldsymbol{W}^{K_R})(\boldsymbol{P} \boldsymbol{W}^{Q_R})^\top = \Re(\boldsymbol{L}_q),$$

$$\tilde{\boldsymbol{P}}^I = \frac{1}{\sqrt{d}} (\boldsymbol{P} \boldsymbol{W}^{K_I})(\boldsymbol{P} \boldsymbol{W}^{Q_I})^\top = \Im(\boldsymbol{L}_q),$$

*then $\boldsymbol{P}$ is out-adjacency-identifying.*

*Proof.* Note that if there exist $\boldsymbol{W}^{K_R}, \boldsymbol{W}^{Q_R}, \boldsymbol{W}^{K_I}, \boldsymbol{W}^{Q_I} \in \mathbb{R}^{d \times d}$ such that

$$\tilde{\boldsymbol{P}}^R = \Re(\boldsymbol{L}_q),$$
$$\tilde{\boldsymbol{P}}^I = \Im(\boldsymbol{L}_q),$$

there also exists $\boldsymbol{W}_*^{K_R}, \boldsymbol{W}_*^{Q_R}, \boldsymbol{W}_*^{K_I}, \boldsymbol{W}_*^{Q_I} \in \mathbb{R}^{d \times d}$ such that

$$\tilde{\boldsymbol{P}}^R = -\Re(\boldsymbol{L}_q),$$
$$\tilde{\boldsymbol{P}}^I = -\Im(\boldsymbol{L}_q).$$

The negative magnetic Laplacian is $-\boldsymbol{L}_q = \boldsymbol{A}_s \odot e^{i\Theta^{(q)}} - \boldsymbol{D}_s$. Note that since $\boldsymbol{D}_s$ is a diagonal matrix, it does not affect the off-diagonal coefficients of $-\boldsymbol{L}_q$. Denote $c := \cos(2\pi q)$ and $s := \sin(2\pi q)$, and consider the different cases.

1. $(i,j) \in E(G)$: $-\boldsymbol{L}_{q,ij} = \boldsymbol{A}_{s,ij} e^{i2\pi q} = e^{i2\pi q}$. Therefore, $-\Re(\boldsymbol{L}_q)_{ij} = c$ and $-\Im(\boldsymbol{L}_q)_{ij} = s$. In that case, $\tilde{\boldsymbol{P}}_{ij} = c + \frac{2-c}{s}s = 2$.

2. $(j,i) \in E(G)$: $-\boldsymbol{L}_{q,ij} = \boldsymbol{A}_{s,ij} e^{-i2\pi q} = e^{-i2\pi q}$. Therefore, $-\Re(\boldsymbol{L}_q)_{ij} = c$ and $-\Im(\boldsymbol{L}_q)_{ij} = -s$. In that case, $\tilde{\boldsymbol{P}}_{ij} = c - \frac{2-c}{s}s = 2(c-1) < 0$.

3. $(i,j), (j,i) \notin E(G)$: $-\boldsymbol{L}_{q,ij} = 0 = -\Re(\boldsymbol{L}_q)_{ij} = -\Im(\boldsymbol{L}_q)_{ij}$. In that case, $\tilde{\boldsymbol{P}}_{ij} = 0$.

4. Since we consider DAGs, the case where $(i,j) \in E(G)$ and $(j,i) \in E(G)$, i.e., $G$ contains a 2-node cycle, never occurs.

The maximum over a row of $\tilde{\boldsymbol{P}}$ is therefore 2, and we have that

$$\tilde{\boldsymbol{P}}_{ij} = \max_k \tilde{\boldsymbol{P}}_{ik} \iff \boldsymbol{A}_{ij} = 1,$$

which concludes the proof. $\square$

The following uses Lemma 2 as a sufficient condition for identifying out-adjacency.

**Lemma 3.** *Let $G$ be a directed graph with magnetic Laplacian $\boldsymbol{L}_q$, with eigendecomposition $\boldsymbol{L}_q = \boldsymbol{U}\boldsymbol{\Lambda}\boldsymbol{U}^*$ and let $\boldsymbol{X} + i\boldsymbol{Y} = \boldsymbol{U}\boldsymbol{\Lambda}^{1/2}$. Then for any permutation matrices $\boldsymbol{M}_X, \boldsymbol{M}_Y \in \mathbb{R}^{n \times n}$, the matrix $\boldsymbol{P} = [\boldsymbol{X}\boldsymbol{M}_X, \boldsymbol{Y}\boldsymbol{M}_Y] \in \mathbb{R}^{n \times 2n}$ is out-adjacency-identifying.*

*Proof.* The idea is to find a pair of suitable bilinear projections to invoke Lemma 2. Note that since $\boldsymbol{M}_X$ and $\boldsymbol{M}_Y$ are permutation matrices, $\boldsymbol{M}_X\boldsymbol{M}_X^\top = \boldsymbol{M}_Y\boldsymbol{M}_Y^\top = \boldsymbol{I}_n$. For $(\boldsymbol{W}^{K_R}, \boldsymbol{W}^{Q_R})$, we choose

$$\boldsymbol{W}^{K_R} = \sqrt{d}\boldsymbol{I}_{2n},$$
$$\boldsymbol{W}^{Q_R} = \boldsymbol{I}_{2n}.$$

Therefore, $\frac{1}{\sqrt{d}}(\boldsymbol{P}\boldsymbol{W}^{K_R})(\boldsymbol{P}\boldsymbol{W}^{Q_R})^\top = \boldsymbol{X}\boldsymbol{M}_X\boldsymbol{M}_X^\top\boldsymbol{X}^\top + \boldsymbol{Y}\boldsymbol{M}_Y\boldsymbol{M}_Y^\top\boldsymbol{Y}^\top = \boldsymbol{X}\boldsymbol{X}^\top + \boldsymbol{Y}\boldsymbol{Y}^\top = \Re(\boldsymbol{L}_q)$. For $(\boldsymbol{W}^{K_I}, \boldsymbol{W}^{Q_I})$, we choose

$$\boldsymbol{W}^{K_I} = \sqrt{d}\begin{bmatrix} \boldsymbol{0} & \boldsymbol{M}_X^\top \\ \boldsymbol{M}_Y^\top & \boldsymbol{0}, \end{bmatrix},$$
$$\boldsymbol{W}^{Q_I} = \begin{bmatrix} \boldsymbol{M}_X^\top & \boldsymbol{0} \\ \boldsymbol{0} & -\boldsymbol{M}_Y^\top \end{bmatrix}.$$

In that case, $\frac{1}{\sqrt{d}}(\boldsymbol{P}\boldsymbol{W}^{K_I})(\boldsymbol{P}\boldsymbol{W}^{Q_I})^\top = [\boldsymbol{Y}\boldsymbol{M}_Y\boldsymbol{M}_Y^\top, \boldsymbol{X}\boldsymbol{M}_X\boldsymbol{M}_X^\top][\boldsymbol{X}\boldsymbol{M}_X\boldsymbol{M}_X^\top, -\boldsymbol{Y}\boldsymbol{M}_Y\boldsymbol{M}_Y^\top]^\top = [\boldsymbol{Y}, \boldsymbol{X}][\boldsymbol{X}, -\boldsymbol{Y}]^\top = \boldsymbol{Y}\boldsymbol{X}^\top - \boldsymbol{X}\boldsymbol{Y}^\top = \Im(\boldsymbol{L}_q)$.

Applying Lemma 2 concludes the proof. $\square$

We are now ready to prove the main theorem.

**Theorem 4.** *Theorem 1, restated) The mLPE is sufficiently out-adjacency-identifying.*

*Proof.* Let us start by recalling some notations. Let $G$ be a $n$-order directed acyclic graph, with magnetic Laplacian $\boldsymbol{L}_q = \boldsymbol{U}\boldsymbol{\Lambda}\boldsymbol{U}^*$. Further write $\boldsymbol{U}\boldsymbol{\Lambda}^{1/2} = \boldsymbol{X} + i\boldsymbol{Y}$, and denote $\boldsymbol{U}^R = \Re(\boldsymbol{U}) \in \mathbb{R}^n$ the real part of $\boldsymbol{U}$ and $\boldsymbol{U}^I = \Im(\boldsymbol{U}) \in \mathbb{R}^n$ the imaginary part of $\boldsymbol{U}$. In what follows, $j$ refers to a column of $\boldsymbol{U}^R$ or $\boldsymbol{U}^I$, and $i$ refers to the index of a node. We denote $u_{ij}^R$ the $i$-th component of $j$-th eigenvector of $\boldsymbol{U}^R$, and similarly for $u_{ij}^I$.

We divide up the $d$-dimensional space of the mLPE into two $\frac{d}{2}$-dimensional sub-spaces, $\mathrm{adj}_{Re}$ and $\mathrm{adj}_{Im}$, i.e. we write, for a node $v_i$,

$$\mathrm{mLPE}(v) = [\mathrm{adj}_{Re}(v_i), \mathrm{adj}_{Im}(v_i)] \in \mathbb{R}^d,$$

where

$$\mathrm{adj}_{Re}(v_i) = \rho_{Re}\left(\sum_{j=1}^n \phi_{Re}(u_{ij}^R, u_{ij}^I, \lambda_j + \epsilon_j)\right),$$

$$\mathrm{adj}_{Im}(v_i) = \rho_{Im}\left(\sum_{j=1}^n \phi_{Im}(u_{ij}^R, u_{ij}^I, \lambda_j + \epsilon_j)\right),$$

and where we have chosen $\rho$ to be a sum over the first dimension of its input, followed by a feed-forward neural network $\rho_{Re}$ and $\rho_{Im}$. For convenience, we also fix an arbitrary ordering over the nodes in $V(G)$ and define

$$\boldsymbol{Q}^R := \begin{bmatrix} \mathrm{adj}_{Re}(v_1) \\ \vdots \\ \mathrm{adj}_{Re}(v_n) \end{bmatrix}$$

$$\boldsymbol{Q}^I := \begin{bmatrix} \mathrm{adj}_{Im}(v_1) \\ \vdots \\ \mathrm{adj}_{Im}(v_n), \end{bmatrix}$$

where $v_i$ is the $i$-th node in our ordering. Note that both $\mathrm{adj}_{Re}(v_i)$ and $\mathrm{adj}_{Im}(v_i)$ are DeepSets over the multiset

$$M_i := \{\!\{(u_{ij}^R, u_{ij}^I, \lambda_j + \epsilon_j)\}\!\}.$$

To prove that $[\boldsymbol{Q}^R, \boldsymbol{Q}^I]$ is sufficiently out-adjacency-identifying, we show that $[\boldsymbol{Q}^R, \boldsymbol{Q}^I]$ approximates $[\boldsymbol{X}\boldsymbol{M_X}, \boldsymbol{Y}\boldsymbol{M_Y}]$ arbitrarily close, for $\boldsymbol{M}_X, \boldsymbol{M}_Y$ some permutation matrices. To that extent, we demonstrate how $\mathrm{adj}_{Re}(v_i)$ (resp. $\mathrm{adj}_{Im}(v_i)$) approximates $\boldsymbol{X}_i$ (respectively $\boldsymbol{Y}_i$) arbitrarily close for all $i$. Since $\mathrm{adj}_{Re}$ and $\mathrm{adj}_{Im}$ are DeepSets, they can approximate any continuous permutation-invariant function over a compact set over the real numbers. Therefore, it remains to show that there exists a permutation invariant function $f^R$ (respectively $f^I$) such that $f^R(M_i) = \boldsymbol{X}_i\boldsymbol{M_X}$ (respectively $f^I(M_i) = \boldsymbol{Y}_i\boldsymbol{M_Y}$) for all $i$ and some permutation matrix $\boldsymbol{M_X}$ (respectively $\boldsymbol{M_Y}$). To this end, note that the magnetic Laplacian graph $G$ of order $n$ has at most $n$ distinct eigenvalues. Hence, we can choose $\epsilon_j$ such that $\lambda_j + \epsilon_j$ is unique, for all $j$. In particular, let

$$\epsilon_j = j \cdot \delta,$$

where we choose $\delta < \min_{l,o}|\lambda_l - \lambda_o| > 0$, i.e., the smallest non-zero difference between two eigenvalues. Now let

$$f^R(M_i) := \left[u_{is_1}^R \cdot \sqrt{\lambda_1 + s_1 \cdot \delta}, \quad \ldots \quad , u_{is_n}^R \cdot \sqrt{\lambda_n + t_n \cdot \delta}\right],$$

$$f^I(M_i) := \left[u_{it_1}^I \cdot \sqrt{\lambda_1 + t_1 \cdot \delta}, \quad \ldots \quad , u_{it_n}^I \cdot \sqrt{\lambda_n + t_n \cdot \delta}\right],$$

where the $\{s_k\}_{k=1}^n$ (respectively $\{t_k\}_{k=1}^n$) are indices in $[n]$ such that the $\lambda_1 + s_k \cdot \delta$ (respectively $\lambda_1 + t_k \cdot \delta$) are sorted in ascending order. This ordering is reflected in some permutation matrix $\boldsymbol{M_X}$ (respectively $\boldsymbol{M_Y}$).

Then, note that, since $\Lambda^{1/2}$ a is real-valued, diagonal matrix, $\boldsymbol{X} = \Re(\boldsymbol{U}\Lambda^{1/2}) = \Re(\boldsymbol{U})\Lambda^{1/2} = \boldsymbol{U}^R\Lambda^{1/2}$, similarly $\boldsymbol{Y} = \Im(\boldsymbol{U}\Lambda^{1/2}) = \Im(\boldsymbol{U})\Lambda^{1/2} = \boldsymbol{U}^I\Lambda^{1/2}$. The $i$-th row of $\boldsymbol{U}^R\Lambda^{1/2}$ is equal to

$$\boldsymbol{X}_i = (\boldsymbol{U}^R\Lambda^{1/2})_i = [u_{i1}^R \cdot \sqrt{\lambda_1}, \quad \ldots \quad , u_{in}^R \cdot \sqrt{\lambda_n}],$$

and, similarly,

$$\boldsymbol{Y}_i = (\boldsymbol{U}^I \Lambda^{1/2})_i = [u_{i1}^I \cdot \sqrt{\lambda_1}, \quad \ldots \quad , u_{in}^I \cdot \sqrt{\lambda_n}].$$

Since we can choose $\delta$ arbitrarily small, we can choose it such that:

$$\|f^R(M_i) - X_i\| < \gamma_R,$$
$$\|f^I(M_i) - Y_i\| < \gamma_I,$$

for any $\gamma_R, \gamma_I > 0$. Since $f^R$ and $f^I$ are continuous permutation-invariant functions over a compact set, a Deepset can approximate them arbitrarily close, and as a result, we have

$$\|\text{adj}_{Re}(v_i) - \boldsymbol{X}_i \boldsymbol{M}_X\| < \gamma_R,$$
$$\|\text{adj}_{Im}(v_i) - \boldsymbol{Y}_i \boldsymbol{M}_Y\| < \gamma_I,$$

for all $i$ and arbitrarily small $\gamma^R, \gamma^I$. In matrix form, it gives

$$\|\boldsymbol{Q}^R - \boldsymbol{X} \boldsymbol{M}_X\| < \gamma_R,$$
$$\|\boldsymbol{Q}^I - \boldsymbol{Y} \boldsymbol{M}_Y\| < \gamma_I,$$

and, further

$$\|[\boldsymbol{Q}^R, \boldsymbol{Q}^I] - [\boldsymbol{X} \boldsymbol{M}_X, \boldsymbol{Y} \boldsymbol{M}_Y]\| < \gamma,$$

where $\gamma = \max(\gamma_R, \gamma_I)$. We know by Lemma 3 that $[\boldsymbol{X} \boldsymbol{M}_X, \boldsymbol{Y} \boldsymbol{M}_Y]$ is out-adjacency-identifying. Since the mLPE can approximate it arbitrarily close, we can, in turn, conclude that mLPE is sufficiently out-adjacency-identifying. $\qquad\square$

## D   IMPLEMENTATION DETAILS

Here, we outline implementation details.

### D.1   AUTOENCODER

**Number of eigenvectors**   Our definition of the LPE and mLPE assumes that we have access to all $n$ eigenvectors. However, this might be impractical for large graphs. In our implementation, we only take the $k$ smallest eigenvalues and their corresponding eigenvectors. In practice, $\phi$ is therefore instantiated as

$$\phi(\boldsymbol{U}_{:,:k}, \boldsymbol{\Lambda}_{:k}) := \Big[ \phi\big(\boldsymbol{U}_{1,:k} \,\|\, \boldsymbol{\Lambda}_{:k} + \epsilon\big), \ldots, \phi\big(\boldsymbol{U}_{n,:k} \,\|\, \boldsymbol{\Lambda}_{:k} + \epsilon\big) \Big] \in \mathbb{R}^{n \times k \times d},$$

where $\boldsymbol{U}_{:,:k}$ denotes the $k$ first columns of $\boldsymbol{U}$, $\boldsymbol{\Lambda}_{:k}$ the $k$ first eigenvalues, $\boldsymbol{U}_{i,:k}$ the $k$ first columns of $i$-th row of $\boldsymbol{U}$.

**Sign invariance**   The eigenvectors of the graph Laplacian are not unique, e.g., if $\boldsymbol{v}$ is an eigenvector, then so is $-\boldsymbol{v}$ (Lim et al., 2023). Hence, GNNs should ideally be *sign invariant*, i.e., represent $\boldsymbol{v}$ and $-\boldsymbol{v}$ identically. To account for this symmetry, we adopt a SignNet-like mechanism (Lim et al., 2023), which randomly flips the signs of eigenvectors during training.

**Decoder DeepSet**   Recall the definition of our edge decoder,

$$\tilde{\boldsymbol{Z}} := \frac{1}{\sqrt{d}} \boldsymbol{Z} \boldsymbol{W}^Q (\boldsymbol{Z} \boldsymbol{W}^K)^\top$$
$$\hat{\boldsymbol{A}} := \sigma(\text{DeepSet}(\tilde{\boldsymbol{Z}})), \quad \hat{\boldsymbol{A}} \in \mathbb{R}^{n \times n}.$$

Here, we describe the architecture of the DeepSet. Let

$$\boldsymbol{C} := \frac{1}{n} \sum_{i=1}^{n} (\tilde{\boldsymbol{Z}} \boldsymbol{W}^C + \boldsymbol{b}^C)_{:,i} \in \mathbb{R}^{n \times d_{DS}}$$

$$\text{DeepSet}(\tilde{\boldsymbol{Z}}) := \boldsymbol{W}^{\text{out}} \Big( \text{GeLU} \Big( \tilde{\boldsymbol{Z}} \boldsymbol{W}^{\text{in}} + \boldsymbol{C} \boldsymbol{W}^{\text{ctx}} + \boldsymbol{b}^{\text{in}} \Big) \Big) + \boldsymbol{b}^{\text{out}},$$

where

- $W^C \in \mathbb{R}^{1 \times d_{DS}}$ and $b^C \in \mathbb{R}^{d_{DS}}$ are learnable parameters; the summation is taken over the projected input rows.

- $W^{\mathrm{in}} \in \mathbb{R}^{1 \times d_{DS}}$, $W^{\mathrm{ctx}} \in \mathbb{R}^{d_{DS} \times d_{DS}}$, and $b^{\mathrm{in}} \in \mathbb{R}^{d_{DS}}$ are learnable parameters. We view $\tilde{Z}$ as a tensor in $\mathbb{R}^{d_{DS} \times d_{DS} \times 1}$, yielding a projection in $\mathbb{R}^{n \times n \times d_{DS}}$, with $C W^{\mathrm{ctx}}$ broadcast across columns.

- $W^{\mathrm{out}} \in \mathbb{R}^{d_{DS} \times 1}$ and $b^{\mathrm{out}} \in \mathbb{R}$ are the final learnable parameters.

**On the problem of structurally equivalent nodes**    Our encoder assigns latent representations to nodes. To avoid handling permutations in the decoder and the loss function, we preserve the nodes' ordering when feeding graphs to the autoencoder. Consequently, a reconstructed edge is considered correct by the loss function only if it exactly matches an edge in the original graph. However, in some situations, this assumption becomes problematic.

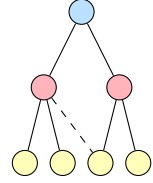

Consider the case illustrated in Figure 4. Structurally equivalent nodes are colored identically; they belong to the same *orbit*. Structurally equivalent nodes also correspond to identical rows in the eigenvector matrix $U$, which in turn means that $\phi$ will assign them identical representations. Since GNNs aggregate information based on neighborhoods, such nodes inevitably receive identical embeddings from $\rho$ as well. For example, the encoder assigns the same representation to all four yellow nodes. The same applies to the two red nodes, which also form an orbit. Consequently, when decoding edges between yellow and red nodes, the decoder should treat a yellow node as equally likely to connect to either red node. Although both predictions are structurally correct, the preserved node ordering forces the loss function to mark one of them as incorrect. Similar observations were made in (Laabid et al., 2025; Morris et al., 2024).

Figure 4: Example of failure case for the decoder. Solid lines denote original edges, while the dashed line denotes a structurally correct prediction that will be considered as incorrect by the loss function.

We experimentally found that this would lead to poor reconstruction performance, particularly in trees, where cases similar to Figure 4 are common. Therefore, we designed a principled way to perturb the input of nodes belonging to non-trivial orbits (i.e., non-singleton orbits) to make them distinguishable in the latent space and ease the decoder's task.

During preprocessing, we identify non-trivial orbits using the final node colorings from the 1-WL algorithm. Although 1-WL does not always distinguish non-equivalent nodes, prior work (Morris et al., 2024) suggests that identical 1-WL colorings serve as a good practical solution to detect structural equivalence. Before feeding eigenvectors to $\phi$, we slightly perturb the rows of $U$ corresponding to nodes in non-trivial orbits by applying a low-magnitude Gaussian modulation. Formally,

$$U_{\mathrm{perturb}} = (1 + \eta M \epsilon) U,$$

where $\eta$ is a small modulation coefficient (typically $\eta = 0.005$), $M \in \mathbb{R}^{n \times n}$ is a mask with rows equal to 1 if a node belongs to a non-trivial orbit and 0 otherwise, and $\epsilon$ is a matrix whose rows are independent Gaussian random vectors.

In simple terms, this modulation injects Gaussian noise into the rows of $U$ corresponding to nodes in non-trivial orbits. Since nodes in the same orbit have different inputs to $\phi$, the corresponding latents will be slightly different. This provides a simple, parsimonious way to make structurally equivalent nodes distinguishable in the latent space. In particular, we found it to yield much better reconstruction results than previously proposed symmetry-breaking methods, such as assigning node identifiers (Bechler-Speicher et al., 2025; You et al., 2019).

### D.2    DIT

**Self-conditioning**    We implement self-conditioning (Chen et al., 2023a) in all our experiments. Self-conditioning provides the denoiser at step $t$ with its own prediction from the previous step as an additional input. During training, this input is dropped with probability 0.5. When conditioning is active, the model first produces a prediction without conditioning, which is then reused as input for a second pass. Gradients are not backpropagated through the first prediction. We find that this approach consistently improves the quality of our results.

**Logit-normal time step sampling**   Following Esser et al. (2024), instead of sampling diffusion times $t$ uniformly from $[0, 1]$, we draw them from a logit-normal distribution. Formally,

$$\text{Logit}(t) = \log\left(\tfrac{t}{1-t}\right) \sim \mathcal{N}(\mu, \sigma^2), \quad t \in (0, 1).$$

In practice, this is implemented by sampling

$$z \sim \mathcal{N}(\mu, \sigma^2), \qquad t = \frac{1}{1 + e^{-z}}.$$

The resulting density is

$$p(t) \propto \frac{1}{t(1-t)} \exp\left(-\frac{(\text{Logit}(t) - \mu)^2}{2\sigma^2}\right),$$

which biases training towards intermediate time steps, the most informative region for flow matching models.

## E   EXPERIMENTAL DETAILS

Here, we outline details on the experiments.

### E.1   DATASET DETAILS

#### E.1.1   SYNTHETIC DATASETS

**Description**   We experiment on three synthetic datasets: EXTENDED TREE, EXTENDED PLANAR, and EGO. The first two extend the benchmarks introduced in Martinkus et al. (2022), namely TREE and PLANAR, which originally consisted of random trees and planar graphs with a fixed size of 64 nodes. These original datasets contained only 200 samples, split into 128/32/40 for train/validation/test, which is too small and leads to high variance during evaluation. Following Krimmel et al. (2025), we extend these benchmarks to 8,192 training samples and 256 samples each for validation and test. The EGO dataset, proposed in You et al. (2018), consists of 757 3-hop ego networks extracted from the Citeseer network, with up to 400 nodes.

**Metrics**   On synthetic graphs, we evaluate our method using the *Maximum Mean Discrepancy* (MMD) metric, as introduced in Martinkus et al. (2022). For each setting, we compute graph statistics on a set of generated samples and the test set, embed them via a kernel, and compute the MMD between the resulting distributions. We report MMD scores for the degree distribution (**Deg.**), counts of non-isomorphic four-node orbits (**Orbit**), clustering coefficient (**Cluster.**), and Laplacian spectrum (**Spec.**). For the EXTENDED TREE and EXTENDED PLANAR datasets, we also report the proportion of valid, unique, and novel graphs (**V.U.N.**), where validity is defined by the dataset constraints (e.g., planarity or tree structure).

#### E.1.2   MOLECULAR DATASETS

**Description**   We next evaluate our method on the Moses benchmark (Polykovskiy et al., 2020), derived from the ZINC Clean Leads collection (Sterling and Irwin, 2015), which comprises molecules with 8 to 27 heavy atoms, filtered according to specific criteria. We also consider the Guacamol benchmark (Brown et al., 2019), based on the ChEMBL 24 database (Mendez et al., 2019). This dataset comprises synthesized molecules tested against biological targets, ranging in size from 2 to 88 heavy atoms. Consistent with prior work (Vignac et al., 2023; Siraudin et al., 2024), the GUACAMOL dataset undergoes a filtering procedure: SMILES strings are converted into graphs and then mapped back to SMILES, discarding molecules for which this conversion fails. We utilize the implementation presented in Vignac et al. (2023) for this process. The standard dataset splits are used.

**Metrics**   For both datasets, we evaluate validity (**Valid**), uniqueness (**Unique**), and novelty (**Novel**). Following Qin et al. (2025), for GUACAMOL we instead report the percentages of valid (**V.**), valid and unique (**V.U.**), and valid, unique, and novel (**V.U.N.**) samples. Each benchmark also includes specific metrics: MOSES compares generated molecules to a scaffold test set, reporting the **Fréchet ChemNet Distance (FCD)**, the **Similarity to the Nearest Neighbor (SNN)**—the average Tanimoto similarity

Table 6: Hyperparameters of LG-VAE on the different datasets.

| Dataset | Planar | Tree | Ego | Moses | GuacaMol | TPU Tile |
|---|---|---|---|---|---|---|
| Number of eigenvectors | 16 | 64 | 64 | 16 | 32 | 64 |
| Latent dim. $d$ | 16 | 24 | 24 | 24 | 24 | 24 |
| Modulation magnitude | 0 | 0.005 | 0.005 | 0.005 | 0.05 | 0.05 |
| Number of layers $\phi$ | 2 | 2 | 2 | 2 | 2 | 2 |
| Embedding dim $\phi$ | 256 | 256 | 256 | 256 | 384 | 256 |
| Number of layers $\rho$ | 16 | 16 | 16 | 16 | 16 | 12 |
| Embedding dim $\rho$ | 256 | 256 | 256 | 256 | 384 | 256 |
| Decoder DeepSet embedding dim | 16 | 16 | 16 | 24 | 24 | 24 |

between fingerprints of generated molecules and those of a reference set—and **Scaffold similarity (Scaf)**, which measures the frequency match of Bemis–Murcko scaffolds between generated and reference sets. Finally, **Filters** indicates the percentage of generated molecules passing the dataset's construction filters. GuacaMol reports two metrics: the **Fréchet ChemNet Distance (FCD)** and the **KL divergence (KL)** between distributions of physicochemical descriptors in the training and generated sets.

### E.1.3 DAG dataset

**Description** We evaluate our DAG generation variant using the TPU Tiles dataset (Phothilimthana et al., 2023a). It consists of computational graphs extracted from Tensor Processing Unit workloads. We split the dataset into 5 040 training, 630 validation, and 631 test graphs.

**Metrics** We compute MMD over statistics analogous to those used for undirected graphs, replacing the degree distribution with incoming (**In Deg.**) and outgoing (**Out Deg.**) degree distributions, and adding features from the graph wavelet transform (**Wave.**). Samples are considered valid if they satisfy the DAG property.

### E.2 Resources

All our models, both autoencoders and DiTs, are trained on two Nvidia L40 GPUs with 40GB VRAM. All models were sampled using a single Nvidia L40 GPU, and we optimized the batch size to fully utilise its memory.

### E.3 Training details

All our autoencoders are trained using the AdamW optimizer, a weight decay of $1e^{-4}$, a KL coefficient of $1e^{-6}$, and a cosine learning rate schedule. We train our DiTs using a constant learning rate of $2e^{-4}$, and maintain Exponential Moving Average (EMA) weights for evaluation.

### E.4 Hyperparameters

**Autoencoder** We use a different number of eigenvectors depending on the dataset, as well as a different value for $d$. Our hyperparameters are listed in Table 6.

**DiT** We experiment using three variants of the DiT: two variants coming from the original DiT paper, DiT Small (DiT-S) and Base (DiT-B), and one smaller variant, coined DiT Tiny (DiT-T), since some of our datasets are smaller than those used in image generation. We detail their hyperparameters in Table 7.

### E.5 Additional experimental results

**Reconstruction performance** See Table 8 for results on reconstruction performance. Note that reconstruction performance on Ego is noticeably lower than on the other datasets. This can be attributed to two factors: the small dataset size (only 606 training samples) and the inclusion of the validation set within the training data, which limits reliable assessment of generalization during training. Nevertheless, Ego

Table 7: Hyperparameters of DiT variants used in our experiments.

| Variant | DiT-T | DiT-S | DiT-B |
|---|---|---|---|
| Number of layers | 6 | 12 | 12 |
| Number of heads | 6 | 6 | 12 |
| Embedding dimension | 384 | 384 | 768 |
| Number of parameters | 16.2 M | 32.2 M | 128.4 M |

does not impose strong structural constraints, and the **Edge. Acc.** remains high (0.9957). As a result, the latent diffusion model can still be trained and generate samples that statistically resemble the training distribution.

Table 8: Reconstruction accuracy of our autoencoder across datasets. **Edge Sample Acc.** and **Node Sample Acc.** denote the proportion of graphs in which, all edges and all nodes, respectively, are correctly reconstructed. For datasets without node labels, **Edge Sample Acc.** coincides with **Sample Acc.**.

| Datasets | Edge Sample Acc. | Node Sample Acc. | Sample Acc. |
|---|---|---|---|
| Extended Planar | 0.9961 | – | 0.9961 |
| Extended Tree | 0.9844 | – | 0.9844 |
| Ego | 0.0861 | – | 0.0861 |
| Moses | 0.9967 | 1.0 | 0.9967 |
| GuacaMol | 0.9973 | 1.0 | 0.9973 |
| TPU Tile | 0.9493 | 1.0 | 0.9493 |

**Ablation on the number of eigenvectors $k$**    We conducted an ablation on the number of eigenvectors used in LG-VAE to better assess the influence of this hyperparameter on reconstruction and generative performance. To this end, we ran experiments on Extended Tree and Extended Planar with $k \in [4, 8, 16, 32, 64]$. Our results are presented in Tables 9 and 10.

On EXTENDED PLANAR, the number of eigenvectors $k$ has only a minor influence on reconstruction accuracy, although MMD metrics worsen for small $k$, consistent with the loss of high-frequency structural information. In contrast, on EXTENDED TREE, $k$ has a much stronger impact on reconstruction quality; interestingly, the best results are obtained with only 25–50% of the eigenvectors, showing that including high-frequencies do not necessarily yield better performance.

Table 9: **Ablation study** : number of eigenvectors on the EXTENDED PLANAR dataset.

| Number of eigenvectors | Sample Acc. | Deg. ($\downarrow$) | Orbit ($\downarrow$) | Cluster. ($\downarrow$) | Spec. ($\downarrow$) | V.U.N. ($\uparrow$) |
|---|---|---|---|---|---|---|
| 4 | 0.9922 | $0.3_{\pm 0.0}$ | $0.7_{\pm 0.2}$ | $13.4_{\pm 2.2}$ | $1.2_{\pm 0.3}$ | $\mathbf{99.4}_{\pm 0.7}$ |
| 8 | 0.9844 | $0.3_{\pm 0.2}$ | $1.0_{\pm 0.3}$ | $12.9_{\pm 3.2}$ | $\mathbf{1.0}_{\pm 0.1}$ | $99.9_{\pm 0.1}$ |
| 16 | $\mathbf{0.9961}$ | $0.3_{\pm 0.1}$ | $0.5_{\pm 0.2}$ | $10.1_{\pm 3.1}$ | $\mathbf{1.0}_{\pm 0.2}$ | $99.0_{\pm 0.2}$ |
| 32 | 0.9844 | $\mathbf{0.2}_{\pm 0.0}$ | $\mathbf{0.1}_{\pm 0.1}$ | $\mathbf{7.6}_{\pm 1.2}$ | $1.1_{\pm 0.1}$ | $99.1_{\pm 0.3}$ |
| 64 | 0.9961 | $0.3_{\pm 0.1}$ | $0.5_{\pm 0.4}$ | $12.8_{\pm 2.3}$ | $1.1_{\pm 0.1}$ | $99.0_{\pm 0.4}$ |

Table 10: **Ablation study** : number of eigenvectors on the EXTENDED TREE dataset.

| Number of eigenvectors | Sample Acc. | Deg. ($\downarrow$) | Orbit ($\downarrow$) | Cluster. ($\downarrow$) | Spec. ($\downarrow$) | V.U.N. ($\uparrow$) |
|---|---|---|---|---|---|---|
| 4 | 0.4648 | $1.3_{\pm 0.1}$ | $0.1_{\pm 0.0}$ | $\mathbf{0.0}_{\pm 0.0}$ | $6.1_{\pm 0.4}$ | $42.3_{\pm 3.2}$ |
| 8 | 0.4688 | $0.7_{\pm 0.0}$ | $\mathbf{0.0}_{\pm 0.0}$ | $\mathbf{0.0}_{\pm 0.0}$ | $5.1_{\pm 0.6}$ | $44.1_{\pm 1.3}$ |
| 16 | 0.9727 | $\mathbf{0.1}_{\pm 0.0}$ | $\mathbf{0.0}_{\pm 0.0}$ | $\mathbf{0.0}_{\pm 0.0}$ | $1.1_{\pm 0.1}$ | $92.3_{\pm 1.2}$ |
| 32 | $\mathbf{0.9883}$ | $\mathbf{0.1}_{\pm 0.0}$ | $\mathbf{0.0}_{\pm 0.0}$ | $\mathbf{0.0}_{\pm 0.0}$ | $1.7_{\pm 0.2}$ | $\mathbf{92.9}_{\pm 1.3}$ |
| 64 | 0.6680 | $\mathbf{0.1}_{\pm 0.1}$ | $0.1_{\pm 0.0}$ | $\mathbf{0.0}_{\pm 0.0}$ | $1.6_{\pm 0.2}$ | $49.9_{\pm 2.8}$ |

**Experiments on larger planar graphs**   See results in Table 11. To evaluate scalability, we trained LG-VAE and LG-Flow on planar graphs of increasing sizes (128, 256). We denote by PLANAR 128 the dataset of planar graphs with 128 nodes, and similarly for 256 and 512. Recall that EXTENDED PLANAR originally contained graphs with 64 nodes.

LG-Flow preserves high performance as graph size grows, achieving strong V.U.N scores and low MMDs, while its inference time increases sub-quadratically. In contrast, DeFoG's performance degrades sharply with graph size, especially on V.U.N. Its quadratic complexity restricts batch sizes during training (6 for PLANAR 256, 2 for PLANAR 512), leading to slow convergence and long training times. LG-Flow, by contrast, supports substantially larger batches (256 for PLANAR 128 and PLANAR 256, 128 for PLANAR 512). This efficiency yields speedups of $340\times$, $262\times$, and $470\times$ on the corresponding datasets.

Table 11: **Reconstruction and generation on larger planar graphs..** We highlight best results in bold.

| $|V|$ | Model | Sample Acc. | Deg. ($\downarrow$) | Orbit ($\downarrow$) | Cluster. ($\downarrow$) | Spec. ($\downarrow$) | V.U.N. ($\uparrow$) | Time (s) ($\downarrow$) |
|---|---|---|---|---|---|---|---|---|
| 64 | DeFog | – | **0.3**$_{\pm0.1}$ | **0.1**$_{\pm0.1}$ | **9.7**$_{\pm0.3}$ | 1.3$_{\pm0.1}$ | **99.6**$_{\pm0.4}$ | $8.9\times10^2$ |
|  | LG-Flow | 0.9661 | **0.3**$_{\pm0.1}$ | 0.5$_{\pm0.2}$ | 10.1$_{\pm3.1}$ | **1.0**$_{\pm0.2}$ | 99.0$_{\pm0.2}$ | **4.6**$_{\pm0.}$ |
| 128 | DeFog | – | **0.0**$_{\pm0.0}$ | 0.7$_{\pm0.2}$ | **3.1**$_{\pm0.8}$ | 1.4$_{\pm0.2}$ | 91.9$_{\pm1.2}$ | $3.8\times10^3$ |
|  | LG-Flow | 1.0000 | 0.1$_{\pm0.0}$ | **0.3**$_{\pm0.2}$ | 4.4$_{\pm1.1}$ | **0.4**$_{\pm0.1}$ | **99.1**$_{\pm0.5}$ | **11.2**$_{\pm0.}$ |
| 256 | DeFog | – | 0.8$_{\pm0.1}$ | 28.6$_{\pm1.8}$ | 49.6$_{\pm3.1}$ | 2.9$_{\pm0.1}$ | 16.8$_{\pm2.7}$ | $1.5\times10^4$ |
|  | LG-Flow | 0.9727 | **0.2**$_{\pm0.1}$ | **2.0**$_{\pm0.8}$ | **17.1**$_{\pm8.0}$ | **0.2**$_{\pm0.0}$ | **97.9**$_{\pm0.5}$ | **57.2**$_{\pm0.0}$ |
| 512 | DeFog | – | 1.2$_{\pm0.1}$ | 56.5$_{\pm2.9}$ | 114.8$_{\pm7.7}$ | 1.5$_{\pm0.0}$ | 1.4$_{\pm0.8}$ | $6.1\times10^4$ |
|  | LG-Flow | 0.8945 | **0.2**$_{\pm0.1}$ | **1.1**$_{\pm0.4}$ | **20.3**$_{\pm6.3}$ | **0.1**$_{\pm0.0}$ | **84.4**$_{\pm3.2}$ | $\mathbf{1.3\times10^2}$ |

**Additional DAG dataset**   In addition to TPU TILE, we evaluated LG-Flow on a synthetic dataset of DAGs generated using Price's model, the DAG analogue of the Barabási–Albert model. We compare our generation results against Directo in Table 12. Both models were sampled with batch size 256 for a fair evaluation of inference efficiency. LG-Flow performs on par with Directo on this dataset while yielding a $244\times$ inference speed-up.

Table 12: **Generation on DAGs: Price's dataset.**

| Method | Out Deg. | In Deg. | Cluster. | Spec. | Wave. | V.U.N | Time |
|---|---|---|---|---|---|---|---|
| Directo | 2.5$_{\pm0.2}$ | 0.2$_{\pm0.0}$ | 9.1$_{\pm0.6}$ | 2.8$_{\pm0.4}$ | 0.5$_{\pm0.1}$ | 98.9$_{\pm0.3}$ | $1.1\times10^3$ |
| LG-Flow | 2.7$_{\pm0.4}$ | 0.3$_{\pm0.1}$ | 12.6$_{\pm5.5}$ | 5.5$_{\pm4.2}$ | 0.6$_{\pm0.5}$ | 98.8$_{\pm0.7}$ | 4.5$_{\pm0.0}$ |

### E.6   EXPERIMENTAL DETAILS ON BASELINES

For fair inference-time evaluation, we used a fixed batch size for all baselines; when this was not feasible, we tuned the batch size to optimize GPU usage. On EXTENDED PLANAR and EXTENDED TREE, all models were sampled with a batch size of 256. On Ego, discrete diffusion models could not fit the full 151-sample test batch on GPU and were sampled with a batch size of 32. On GuacaMol and MOSES, all models were sampled with a batch size of 1000. On TPU Tile, we sampled both Directo and LG-Flow using a batch size of 40, i.e. the full test batch.

## F   LIMITATIONS

The primary limitation of our approach is the requirement for sufficient data to train the autoencoder to achieve nearly lossless reconstruction. Yet, most existing graph generation datasets are relatively small, as they were primarily designed for diffusion models operating directly in graph space—a computationally costly setting. Since our model scales much more efficiently, it opens the possibility of leveraging substantially larger datasets, which in turn mitigates this limitation. Training on parametric distributions such as Stochastic Block Models also proves difficult, since nodes are stochastically indistinguishable and thus structurally equivalent. Nevertheless, we argue that such datasets offer limited practical value, as these simplistic random models fail to capture the complexity of real-world graph distributions.

