# OpenReview forum: "Principled Latent Diffusion for Graphs via Laplacian Autoencoders"
_ICLR.cc/2026/Conference — Submitted to ICLR 2026_

### Official Review · Reviewer_5ueS · 2025-10-15

**Soundness:** 3
**Presentation:** 4
**Contribution:** 4
**Rating:** 6
**Confidence:** 5

**Summary:**

This article proposes a nearly lossless latent graph diffusion method in view of the high complexity limitations of the existing graph diffusion models due to the discrete paradigm. Specifically, considering the strict requirements of graph data for reconstructing encoder-decoder, the authors proposed a Laplacian autoencoder, which was proven to be able to compress graph data into low-dimensional vectors nearly lossless. Then, the author placed it in the framework of stream matching and trained it using DIT. It is worth noting that this method has also been extended to the research in the field of directed graphs generation.

**Strengths:**

1.I recognize the contribution of this paper in the field of graph generation. Unlike the mature autoencoders in the field of computer vision, the field of graph learning still lacks such an effective lossless encoding method.

2.This paper extends the framework of graph generation to directed graphs, which is of great significance. Previous work mainly focused on molecular data or synthetic undirected graph structure data. Generating directed graphs is equally important.

3.The paper provides theoretical guarantees for the method.

4.The paper is well-written and the introduction of the motivation is very convincing.

**Weaknesses:**

1.I think that compared with the design in graph generation, the author's contribution mainly focuses on a powerful autoencoder. I think this is very important, but there is a lack of sufficient research and innovation on the generation method.

2.Regarding the autoencoder, the author achieved nearly lossless results in the experiment. However, as the author mentioned, their assumption is that the node order of the graph remains unchanged, but this can cause problems in some specific scenarios (tree-like structure). Their processing method is to use wl-test to color the nodes. I think this point is worthy of exploration and analysis. For instance, if appropriate position encodings are assigned to the graph, is there a possibility of a solution?

3.One concern about the autoencoder is that although this method can be extended to large graphs by virtue of its complexity advantage, the number of samples in large graphs is equally scarce. Then, when only a limited number of large graphs are used for training, can lossless compression still be achieved? I suggest that the author could attempt to train on the sbm datasets(you can generate by yourself) using only 10 graphs each containing 10,000 nodes to train and verify the reconstruction performance of the autoencoder on them.


4.Another concern is that this method achieves a lossless effect on the encoder, but it does not have a significant advantage in terms of generation effect. However, the author lacks profound analysis and discussion. Is this due to the error of the generative model or the inherent distribution difference brought about by the training set and the test set?

5.Furthermore, it would be great to be able to see the comparison of the effects of different generation strategies. For instance, the effect differences between ddpm and flow matching, as well as the influence of the selection of different compression dimensions on the reconstruction and generation effects. I hope the author can increase experiments on different diffusion paradigms and, most importantly, compare the impact of choices in different dimensions.

**Questions:**

All my questions are shown in Weakness. I will consider improving my score if the authors can answer my questions.

---

> ### Author Response · Authors · 2025-11-22
> **Answer to reviewer 5ueS**
>
> We thank the reviewer for their detailed feedback and constructive suggestions. Please find our responses below:
>
> - Focus on autoencoder design
>
> This point was introduced in lines 58–70, but we are happy to clarify our positioning. A key advantage of latent diffusion is that it eliminates the need to design entirely new graph diffusion architectures, a process that is both costly and engineering-intensive. Substantial architectural progress has already been made in image diffusion, and we aim to bridge graph generation with this line of work to benefit from those advances.
>
> This shift emphasizes building effective graph autoencoders rather than developing large, specialized graph diffusion models. This is more resource-efficient, since graph autoencoders typically rely on GNNs for encoding, which are comparatively inexpensive to train. Moreover, autoencoders can be tailored to the structure of graph data without requiring direct manipulation of the mathematical foundations of diffusion or flow matching, making the approach more accessible.
>
> - Node identifyability and positional encoding
>
> Regarding node ordering, this is also preserved in discrete graph diffusion models. Shuffling nodes in the latent space would force the decoder to solve a graph-matching problem, which is significantly harder.
> The underlying issue is node identifiability in the latent space. We explored multiple solutions before settling on the approach described in the appendix, including random features and positional encodings such as distances to anchors. The eigenvector modulation we use was the only method that yielded satisfactory results. We agree that this remains an interesting challenge and a valuable direction for future work.
>
> - Experiments on large graphs
>
> We acknowledge a limitation of our method: a sufficient number of samples is necessary to fit the autoencoder. Our attempts to fit the LG-VAE on a dataset such as Point Clouds (see [1]), which contains 26 graphs for training, with up to 5000 nodes, were unsuccessful. This could be due to the strong class imbalance between present and absent edges: in sparse graphs, the number of existing edges grows only linearly with the number of nodes, whereas the number of absent edges grows quadratically. We leave for future work to design a decoder that works better for large networks (< 1000 nodes).
>
> However, to demonstrate the potential of our method to scale to larger graphs than those we considered in our experiments, we designed a dedicated set of experiments, and are currently training models on planar graphs of increasing sizes (128, 256, 512).
>
> [1] Efficient and Scalable Graph Generation through Iterative Local Expansion, Bergmeister et al., ICLR 2024
>
> - Lack of analusis on generative performance
>
> We understand the concern that our approach does not consistently outperform the baselines. It is likely that decoding errors affect some results, especially on validity and VUN metrics.
> The situation differs across datasets. On the synthetic benchmarks (Tree and Planar), performance is already saturated: most models achieve near-perfect VUN and very low MMD scores, leaving little room for improvement. Even with decoding errors, our method remains competitive. We also plan to check whether larger planar graphs provide further differentiation.
> On molecular datasets, we acknowledge that our method does not match DeFoG on some metrics. However, validity and uniqueness are not the only criteria of interest. For example, on GuacaMol, although our VUN is lower than DeFoG and Cometh, we outperform both by a large margin in FCD. This suggests that their higher VUN may come at the cost of lower fidelity to the underlying distribution. A similar pattern appears with DeFoG on MOSES. The lower novelty on MOSES may be due to our higher-capacity architecture, which could lead to more memorization. This issue could be mitigated when using larger training corpora.
>
> - Different generative framework (e.g. DDPM)
>
> We agree that exploring a stochastic framework could be interesting. However, diffusion models and flow matching have been shown to be largely equivalent in their standard formulations (see [1]). Given the training cost of diffusion models and the overall experimental load for this rebuttal, it is unlikely that we will have sufficient time to run these additional experiments. If time permits, we would be glad to include them.
>
> [1] Diffusion Models and Gaussian Flow Matching: Two Sides of the Same Coin, Gao et al., The Fourth Blogpost Track at ICLR 2025.
>
> **If we answered your concerns and question, please consider increasing your score. We are happy to answer any remaining questions.**

---

> > ### Comment · Reviewer_5ueS · 2025-11-26
> > **Response to rebuttal**
> >
> > Thank you for the author's reply. All my concerns about some explanatory issues have been resolved. I understand that due to time constraints, the authors are unable to conduct a more extensive experimental exploration and analysis. Based on the above considerations and the overall assessment of the paper, I still maintain my positive score unchanged. In addition, I encourage the authors to further explore larger-scale graphs and the generation of biological macromolecules, as this will make even greater contributions to this field. In any case, I think this is a good paper, so I gave it a positive score.

---

### Official Review · Reviewer_3gKK · 2025-10-26

**Soundness:** 3
**Presentation:** 2
**Contribution:** 3
**Rating:** 4
**Confidence:** 3

**Summary:**

The paper proposes a latent diffusion framework for graph generation that uses an autoencoder with Laplacian based adjacency identifying positional encodings. The decoder applies bilinear attention style scoring and a row wise DeepSet, and the approach extends to DAGs via a magnetic Laplacian. The authors claim that after freezing the autoencoder, a Diffusion Transformer is trained in latent space with conditional flow matching, which yields mostly linear per node complexity with only the final adjacency decoding quadratic. Experiments report near lossless reconstruction on several datasets, competitive generation quality.

**Strengths:**

- The experiments show competitive reconstruction performance while substantially reducing the runtime.
- Efficient yet near-lossless graph generation is an important topic.

**Weaknesses:**

- Since the model uses only the lowest Laplacian eigenvalues/eigenvectors (rather than the full spectrum), it leans toward global/low-frequency structure and may miss high-frequency details. It might potentially hurt the exact reconstruction of fine structures.
- The approach depends on computing Laplacian eigenvectors for positional encodings, which is costly and can dominate preprocessing on large graphs.
- The promise of near-lossless reconstruction appears to rely on having sufficiently large training corpora of graphs, and performance can degrade on smaller or more complex datasets.

**Questions:**

- Recent graph representation learning works [2] suggest that emphasizing low-frequency components can improve downstream representations (e.g., node classification), so selecting the k smallest eigenpairs is often reasonable in that context. However, for generation, where accurately reconstructing fine structural details could be crucial, restricting to the lowest eigenpairs may suppress high-frequency information. If I am wrong, please correct me.
- In (e.g., line 174), writing $(\phi:\mathbb{R}^2\to\mathbb{R}^d)$ can suggest the input itself is 2-dimensional, whereas the actual input per node is an $(n\times 2)$ table formed by concatenating two vectors row-wise (each row is a 2-vector ($(U_{ij},\lambda_j{+}\epsilon_j))$). Could you explicitly state that $(\phi)$ is applied row-wise and annotate the shapes (input $(n\times 2)$ → output ($n\times d)$) to avoid confusion?
-  In the notation section, node features $X$ are defined as discrete labels. Does the current framework support graphs with dense real-valued node attributes (e.g., Cora dataset $n\times d$ features)?
- Several recent works also build graph learning/generation models from spectral perspectives (e.g., using the smallest (k) eigenpairs). Could the authors discuss the key differences between your design and [1,2]?

[1] "Generating Graphs via Spectral Diffusion." The Thirteenth International Conference on Learning Representations, 2025.

[2] "SDMG: Smoothing Your Diffusion Models for Powerful Graph Representation Learning." The Forty-second International Conference on Machine Learning, 2025.

---

> ### Author Response · Authors · 2025-11-22
> **Answer to reviewer 3gKK**
>
> We thank the reviewer for their thorough evaluation and thoughtful comments. Please find our detailed responses to each point below:
>
> > The approach depends on computing Laplacian eigenvectors for positional encodings, which is costly and can dominate preprocessing on large graphs.
>
> While computing Laplacian eigenvectors can be expensive for very large graphs, we note that this operation has been extensively optimized over the years and can now be performed efficiently in practice, e.g., via highly optimized version of Krylov subspace methods or Locally Optimal Block Preconditioned Conjugate Gradient (+ preconditioner), which is well explored in numerical linear algebra and HPC; see, e.g., Y. Saad, "Numerical Methods for Large Eigenvalue Problems", SIAM Review 2011 or : Andrew V. Knyazev, “Toward the Optimal Preconditioned Eigensolver: Locally Optimal Block Preconditioned Conjugate Gradient Method”. Moreover, it is a one-time preprocessing step, so it incurs a fixed cost rather than affecting the runtime of the model itself.
>
> > The promise of near-lossless reconstruction appears to rely on having sufficiently large training corpora of graphs, and performance can degrade on smaller or more complex datasets.
>
> We acknowledge that our method requires a sufficient number of samples to train the autoencoder. However, our experiments on synthetic graphs show that good reconstruction performance can be achieved with datasets of only a few thousand samples, which is far from a large-scale training setting.
>
> > However, for generation where accurately reconstructing fine structural details could be crucial, restricting to the lowest eigenpairs may suppress high-frequency information.
>
> We acknowledge that, in principle, using the full spectrum would offer maximal expressiveness. However, this is often impractical for large graphs. Our experiments show that we can achieve high reconstruction accuracy even without the highest frequencies. This indicates that we can trade high frequencies for computational efficiency with limited impact on performance.
> We also conducted an ablation study on number of eigenvectors k (see Appendix). Our results show that using the full spectrum is not necessarily optimal, as best results are usually obtained using 25/50 % of the eigenvectors.
>
> - Suggested clarification on the definition of $\phi$
>
> We appreciate the reviewer’s precise comment aimed at improving the clarity of our presentation. However, we would like to note that Lines 179–181 already explain that $\phi$ is applied to the tensor of eigenvectors and eigenvalues along its last dimension. In addition, Line 318 states that $\phi$ is applied row-wise. We will clarify this further in the revised manuscript by explicitly stating that $\phi$ operates on the last dimension of the tensor.
>
> - Does the current framework support graphs with dense real-valued node attributes ?
>
> We indeed focus on categorical attributes in this work. However, there is no reason we could not support real-valued attributes. Since node attributes are predicted from the latents using a simple linear layer, adapting the decoder would be straightforward by replacing the reconstruction loss with a mean squared error. In that setting, graph diffusion models operating directly in graph space (e.g., DeFoG, DiGress) would need separate diffusion processes for discrete features and continuous attributes, as done for instance in MiDi. In contrast, our approach embeds all information in a latent space, so the diffusion model operates on a single continuous modality.
>
> - Discussion on additional references
>
> We thank the reviewer for bringing those two references to our attention. Let us highlight the key differences between our approach and those works.
>
> Regarding [1], the approach is fundamentally different from ours: it generates approximate eigenvalues and eigenvectors and then reconstructs the adjacency matrix through a separate prediction module. In contrast, our approach embeds structural information in a latent by feeding eigenvectors and eigenvalues to the encoder.
>
> Concerning [2], the scope of the paper is different to ours. The authors focus on representation learning using graph diffusion models, while our method is evaluated solely for generation. In contrast to traditional graph diffusion models which target the whole adjacency matrix, they train their model to only reconstruct the low-frequency eigenvectors of graph Laplacian.
> Overall, these methods also rely on the decomposition of the graph Laplacian, but they aim at reconstructing its eigenvectors, whereas we use it to embed structural information into the latent space.
>
> **If we answered your concerns and question, please consider increasing your score. We are happy to answer any remaining questions.**

---

> > ### Comment · Reviewer_3gKK · 2025-11-26
> > **Response to the Rebuttal**
> >
> > Thank you for the author’s response. My questions have been resolved, so I am willing to increase my rating. However, I noticed that the revised manuscript does not yet explicitly address all the **4 questions** I raised, though I trust this can be resolved in the final version. Please revise the manuscript accordingly.

---

### Official Review · Reviewer_d3hS · 2025-10-29

**Soundness:** 1
**Presentation:** 2
**Contribution:** 3
**Rating:** 4
**Confidence:** 3

**Summary:**

ML graph generation models suffer from poor scalability due to quadratic cost of graph representation and hardness to compress such graphs into revertible latent representation. LG-VAE poses itself as a variational autoencoder that uses the graph laplacian spectrum to bring provable reversibility and enable fast graph generation. LG-VAE is shown to generate graphs up to 1000x faster than state-of-the-art while retaining generative quality.

**Strengths:**

- The proposed method brings outstanding improvements under inference cost compared to the state-of-the-art
- The proposed method brings comparable generative performance than the state-of-the-art
- The proposed method covers an extensive set of graphs, including attributed, non-attributed, and directed acyclic graphs

**Weaknesses:**

- The main contribution of the paper is scalable graph generation. However, the experiments mostly cover the generation of small graphs.
- How evaluation is conducted is partly unclear, making the validity of the paper’s result debatable. For instance, it is not clear whether LG-- VAE has been tested with the same batch size as the baselines, a detail that could invalidate a significant part of the paper contribution.
- Experiments on directed graphs seem to be insufficient. First, LG-VAE is tested on one dataset only. Secondly, inference time of the SOTA baseline is inferred by their paper, meaning that it has also been run on a different hardware.
- The experiment section could cover more ablation studies to properly assess the impact of each of the proposed components.
- Minor: methodology figure could be improved for clarity and making LG-VAE’s components easier to understand.

**Questions:**

Altogether, I think the paper makes a reasonable amount of contributions and that the experiments showcase that LG-VAE is a superior choice for graph generation with diffusion models. However, I have concerns about how the evaluation of LG-VAE has been performed. Here are my questions for the authors:

- The main contribution of LG-VAE is scalable generation. However, the experiments mostly cover datasets of small graphs. Could the authors perform experiments on datasets with larger graphs? Proteins [1] could be a good choice also used in several papers in this field. If the authors have preference for any other dataset with large graphs over Proteins would also be fine.
- Have the baselines and LG-VAE tested with the same batch size? If not, what batch sizes have been used in the experiments? This is not clearly explained in the paper content. From what I understand, the reason why LG-VAE is much faster than the SOTA in Table 1 is because the inference has been run with one batch only for LG-VAE but not for the other models. If this is the case could the author measure memory cost and time cost of LG-VAE and the baselines with the same batch sizes?
- The paper would benefit from more ablation studies to assess the impact of each of the components. For instance, experiments to test the impact of the choice of k as the number of used eigenvectors for encoding on computational costs and generation quality. Could the authors provide such experiments?
- Could the authors run the experiments of Table 5 on the same hardware to verify the computation efficiency improvement wrt Directo?
- Could the authors find at least one more DAG dataset to perform experiments on?
- Fu et al. [2] already provided a method for graph diffusion with cost linear in the number of nodes. The paper cites this work but does not compare LG-VAE performance with this model. Could the authors compare LG-VAE computation time and generation quality with the one of HypDiff?

[1] Dobson, P.D. and Doig, A.J., 2003. Distinguishing enzyme structures from non-enzymes without alignments. Journal of molecular biology, 330(4), pp.771-783.
[2] Fu, X., Gao, Y., Wei, Y., Sun, Q., Peng, H., Li, J. and Li, X., 2024. Hyperbolic geometric latent diffusion model for graph generation. arXiv preprint arXiv:2405.03188.

---

> ### Author Response · Authors · 2025-11-22
> **Answer to reviewer d3hS**
>
> We thank the reviewer for their thoughtful feedback and constructive suggestions. Please find our detailed responses below:
>
> > Minor: methodology figure could be improved for clarity and making LG-VAE’s components easier to understand.
>
> We appreciate the reviewer’s comment aimed at improving the clarity of our paper. Could the reviewer indicate which specific aspects of the figure were unclear? Concrete feedback would be extremely valuable to us when revising the figure.
>
> - Large graph generation
>
> First, let us clarify our contribution regarding scalability. Prior methods, such as DiGress and DeFoG, are fundamentally constrained by their quadratic complexity in the number of nodes, which makes them both slow and highly memory-intensive even on small graphs, as evidenced by their generation times on synthetic graphs and molecules. In contrast, our approach enables the use of high-capacity architectures while preserving very fast inference, representing a substantial improvement over existing work.
>
> Regarding larger graphs, we acknowledge a limitation of our method: it requires sufficient training data to fit the autoencoder. Unfortunately, large-graph generation benchmarks are scarce and typically provide too few samples for effective training. For example, the Proteins dataset contains only a few hundred training graphs. Because of this limited scale, our method does not achieve satisfactory reconstruction performance on this dataset. This is likely due to the strong class imbalance between present and absent edges: in sparse graphs, the number of existing edges grows only linearly with the number of nodes, whereas the number of absent edges grows quadratically.
>
> To better assess scalability, we have therefore designed a dedicated set of experiments. We are currently evaluating our approach on planar graphs of increasing size (128, 256, and 512 nodes). For reference, Proteins includes graphs with up to 500 nodes and an average of roughly 250.
>
> - Inference time evaluation and batch size
>
> We acknowledge that our evaluation of inference efficiency lacked precision and, in some cases, rigor. We clarify here how the experiments were conducted. All sampling runs were performed on a single L40 GPU with 48GB of RAM.
>
> For Extended Tree and Planar, we used a batch size of 256 for all models, meaning the complete test set was sampled in a single batch. For Ego, we optimized GPU usage. Since our method is substantially more efficient than discrete diffusion models such as DeFoG, we were able to fit the entire test batch of 151 samples on the GPU. In contrast, as shown in Figure 3.c, DeFoG runs out of memory when its batch size is increased from 32 to 64. The results in Table 3.b are therefore obtained with a batch size of 32.
> While we understand the concern for fair comparison, this does not require using the same batch size for all models. Instead, each architecture should be run under its optimal GPU configuration. If our method is more efficient, this is an advantage and should be reflected in the evaluation.
>
> Concerning Moses, GuacaMol and TPU Tile, we have resampled baselines with equal batch size / optimized GPU usage for a fair evaluation of inference time.
>
> - Ablation study on the number of eigenvectors
>
> To clarify the role of this hyperparameter, we conducted an ablation study on the number of eigenvectors k, with results reported in Appendix E.5. On Planar, k has a limited effect on reconstruction accuracy, while MMD performance degrades for small values of k, as expected from the loss of high-frequency structural information. On Extended Tree, k has a more substantial impact on reconstruction. Still, the best performance is achieved with only 25–50% of the eigenvectors, indicating that increasing k beyond this range does not provide additional benefits.
>
> - Additional DAG dataset
>
> We acknowledge that our evaluation on DAGs includes fewer datasets than in the undirected setting, and we will make every effort to provide an additional DAG experiment as soon as possible.
>
> > Could the authors compare LG-VAE computation time and generation quality with the one of HypDiff?
>
> Although we were aware of this work, its limited performance on challenging benchmarks such as Moses and GuacaMol led us not to include it as a baseline. Given the time required to run experiments across all datasets in our study, it is unlikely that we can provide results on all of them. We will include results at least on the synthetic datasets.
>
> **If we answered your concerns and question, please consider increasing your score. We are happy to answer any remaining questions.**

---

> > ### Comment · Reviewer_d3hS · 2025-11-25
> > **Response to the Rebuttal**
> >
> > I would like to thank the authors for their answers to my questions. I appreciate the addition of the ablation study on the number of used eigenvectors in Appendix E.5.
> >
> > For what concerns large graph generation, the fact that LG-VAE/LG-Flow does not scale at large graph generation due to the training costs is not explicitly mentioned in the main paper as far as I can tell. At line 43 of the revised paper, the authors explicitly say that (LG-VAE) "yields efficient training". Again at line 55 "enable the training of larger and more expressive generative models".
> >
> > (minor) For what concerns Fig 1, I struggle in understanding it as a clear high-level representation of the paper novelties. To me it looks more like a detailed description of the operations that LG-VAE uses.
> >
> > While I see some improvements from the state-of-the-art, I suggest that paper needs to revise its rigor in both its content and evaluation. Further, I see other reviewers raised more concerns to which I tend to be aligned with.
> >
> > Overall I keep my score of 4. That is: marginally below the acceptance threshold. But would not mind if paper is accepted.

---

> > > ### Author Response · Authors · 2025-11-25
> > >
> > > > For what concerns large graph generation, the fact that LG-VAE/LG-Flow does not scale at large graph generation due to the training costs is not explicitly mentioned in the main paper as far as I can tell.
> > >
> > > It appears there has been a misunderstanding. We wish to clarify that the concern you raise does not reflect the explanations provided in our responses to the reviewers and in our general comment. We never state anywhere that training LG-VAE on large graphs is limited by training costs. What we acknowledge is the need for sufficient data to train the variational autoencoder.
> > >
> > > As for l.55, we indeed manage to train larger models. For reference, models like DeFoG are usually capped at ~10M parameters due to their quadratic complexity, while our models can scale gently at sizes up to 100M parameters as indicated in appendix E.4. In the perspective of training on large scale data corpora, it's clearly an advantage for our method, the same way latent diffusion allowed scaling diffusion models for images beyond DDPM.
> > >
> > > Concerning Fig. 1, it is intended to offer clarity on the architecture by illustrating the design and flow of the autoencoder, rather than serving as a standalone high-level summary of the contributions.
> > >
> > > Regarding rigor, we have answered your question regarding fairness in evaluation (also see general comment) by aligning the sampling setup across the models, which was indeed a concern shared amongst some reviewers. If you have other concerns, please point them so that we have chance to answer them.
> > >
> > > Finally, you asked for more experiments, that we are currently trying to run, especially an additional DAG dataset that is on the way. We kindly ask for your patience while we finalize these results.

---

### Official Review · Reviewer_VxLZ · 2025-10-31

**Soundness:** 2
**Presentation:** 3
**Contribution:** 3
**Rating:** 4
**Confidence:** 2

**Summary:**

Graph generation suffers from quadratic complexity in the number of nodes stemming from the adjacency matrix, which, however, contains mostly zeros for most datasets. This hinders the generation of large graphs. The paper proposes a permutation-equivariant autoencoder that maps each node into a fixed-size latent space, leveraging the graph Laplacian spectrum to reduce complexity. The latents are then used to train a generic diffusion model.

**Strengths:**

- Strong generation speedup compared to state-of-the-art baselines.
- Quality-wise, the method achieves the best or second-best performance against the baselines.
- Experiments on a large variety of different types of graphs, both synthetic and real-world, e.g., DAG, molecules, planar, trees.
- Good reconstruction results.

**Weaknesses:**

- Motivation stems from limitations affecting the generation of large graphs, but evaluation only covers standard graph benchmark datasets of the same size as tackled by related work. Memory and time might not be the only limits when generating graphs of larger sizes. It is unclear if quality can be maintained.
- Unclear how comparable the baseline results are on inference time: e.g., how optimal/equal were the used batch sizes and if the same hardware was used. Especially DAG runtimes are inferred from the original authors' paper, which might not have been using the same setup.
- No ablation studies to better highlight the impact of the different components.

Minor:
- Table 2: Sample acc. column: wrong row is highlighted in bold

**Questions:**

- How is the generation quality on larger graphs?
- A sensitivity analysis on the number of eigenvectors used
- What batch size is used, and what hardware is used for inference (appendix mentions 2x L40 for training)?
- How similar is the hw form Directo? Can the experiments be run on the same hardware?

---

> ### Author Response · Authors · 2025-11-22
> **Answer to reviewer VxLZ**
>
> We thank the reviewer for their thoughtful feedback and detailed suggestions. We address each point below:
>
> > Table 2: Sample acc. column: wrong row is highlighted in bold
>
> Thanks for bringing this to our attention. We will correct this small mistake in an updated version of the manuscript. The actual sample accuracy value for the raw GNN encoder is 0.
>
> - How is the generation quality on larger graphs?
>
> We would like to clarify our contribution regarding scalability. Prior methods, such as DiGress and DeFo,G are fundamentally constrained by their quadratic complexity in the number of nodes, which makes them both slow and highly memory-intensive even on small graphs, as reflected in their generation times on synthetic graphs and molecules. In contrast, our approach supports high-capacity architectures while maintaining very fast inference, representing a substantial improvement over existing work.
>
> Regarding larger graphs, we acknowledge a limitation of our method: it requires sufficient training data to fit the autoencoder, a requirement that is not met by the few datasets commonly used for large graph generation. However a few thousand samples already suffice, as shown in the synthetic experiments. Therefore, we are conducting experiments on planar graphs of increasing size (128, 256, and 512 nodes) to assess the scalability of our approach further.
>
> - Sensitivity analysis on the number of eigenvectors
>
> To clarify the role of this hyperparameter, we conducted an ablation study on the number of eigenvectors k, with results reported in Appendix E.5. On Planar, k has a limited effect on reconstruction accuracy, while MMD performance degrades for small values of k, as expected from the loss of high-frequency structural information. On Extended Tree, k has a more substantial impact on reconstruction. Still, the best performance is achieved with only 25–50% of the eigenvectors, indicating that increasing k beyond this range does not provide additional benefits.
>
> - Inference time evaluation and hardware
>
> Our initial inference-efficiency evaluation lacked precision, and we provide a clearer, more rigorous account here. All sampling runs were executed on a single L40 GPU (48GB RAM). For Extended Tree and Planar, all models were run with a batch size of 256, enabling complete test-set sampling in one pass. For Ego, we optimized GPU usage: our method could fit the whole batch of 151 samples, whereas DeFoG ran out of memory beyond a batch size of 32 (Figure 3.c), so its reported runtimes correspond to batch size 32.
> For Moses, GuacaMol, and TPU Tile, we are currently resampling all baselines with equal batch sizes to ensure a fair and comparable evaluation across models.
>
> **If we answered your concerns and question, please consider increasing your score. We are happy to answer any remaining questions**

---

> > ### Comment · Reviewer_VxLZ · 2025-11-25
> >
> > I would like to thank the authors for addressing my questions, providing clarifications, and the extra ablation study on the impact of the number of eigenvectors k. I look forward to the additional experiments on larger graphs.

---

### Author Response · Authors · 2025-11-22
**Answer to all reviewers**

We would like to thank all the reviewers for evaluating our work and providing insightful feedback. Several concerns are shared across reviews, and we are conducting the necessary experiments to address them.

Specifically, we identified three main issues:

- The absence of an ablation study, particularly regarding the influence of the number of eigenvectors k
- The fairness of the inference-time comparison, given that some results were taken from original papers
- The lack of evaluation on large graphs.

To address these points, we will provide the following additions/clarifications :

- We have resampled Directo on the same hardware used for our model (a single L40 GPU), and updated the inference time in Table 5.
- On GuacaMol, we noticed that Cometh, DeFoG and our approach didn’t use the same number of samples for evaluation. We have resampled all three models using the same batch size (1000) and the same number of generated samples (10k). We have also resampled models on MOSES using the same batch size (1000), and for Cometh, the same number of diffusion steps than DeFoG (1000 steps).
- For the Extended Tree and Planar datasets, we had already sampled all baselines on the same hardware with a batch size of 256. For Ego, we optimized GPU usage; due to its computational efficiency, our model is the only one able to accommodate the full test batch of 151 samples on GPU.
- We will include all those details on the experimental setup used for baselines in Appendix.
- We have conducted an ablation study on k for the Extended Planar and Tree datasets; results will appear in the Appendix of the updated version.
- We have answered concerns regarding large-graph generation by performing experiments on Planar graphs of increasing size (128, 256); we will report our findings in the Appendix as well.

We are still actively working on addressing the reviewers’ questions and concerns. We will soon include results for planar graphs of size 512, and we are running DeFoG on these datasets of larger graphs to provide a baseline. We will also try to add an additional DAG dataset, as requested by reviewer d3hS, and, if time permits, include results for another baseline, HypDiff.

**An updated version of the paper has been uploaded, with all modifications highlighted in red.** We remain available for further questions and welcome continued discussion with the reviewers.

On a more practical note, there is some confusion between the LG-VAE (the autoencoder) and the latent flow-matching model. To avoid ambiguity, we will rename the latter LG-Flow in the revised version.

---

### Meta-Review · Area_Chair_2B4H · 2026-01-06

**Summary:**

The reviewers generally agree that the paper proposes a technically solid and timely approach to scalable graph generation via latent diffusion. The main concerns centered on the evaluation on larger graphs, questions about the fairness and clarity of inference-time comparisons, limited ablation studies, and the claims about scalability and near-lossless reconstruction.

**Reviewer Concerns:**

Addressed Concerns:
New experiments on large graphs addressed the scalability concern.The authors clarified the experimemtal settings and baselines where needed.A detailed ablation study was added, showing strong performanceand alleviating concerns about sensitivity to this hyperparameter.An additional DAG experiment was included, strengthening the evaluation.

Remaining concerns:
Some reviewers remain cautious about the near-lossless reconstruction and scalability claims.The method does not consistently outperform baselines on all metrics, and deeper analysis of failure modes and trade-offs was suggested but not fully explored.

**Reviewer Scores:**

Reviewer 3gKK (4→ 6): Indicated willingness to raise their score.

Reviewer 5ueS (6→ 6): Maintained a positive assessment throughout.

Reviewer d3hS (4→ 4): Would likely keep their score unchanged, due to persistent concerns about rigor and presentation.

Reviewer VxLZ (4→ 4 or 6): Likely to increase after the added large-graph experiments, ablations, and clarified runtime evaluation.

---

### Decision · Program_Chairs · 2026-01-26

Reject